# Black-Box Generalization:
# Stability of Zeroth-Order Learning

**Konstantinos E. Nikolakakis**
Yale University
`konstantinos.nikolakakis@yale.edu`

**Farzin Haddadpour**
Yale University
`farzin.haddadpour@yale.edu`

**Dionysios S. Kalogerias**
Yale University
`dionysis.kalogerias@yale.edu`

**Amin Karbasi**
Yale University & Google Research
`amin.karbasi@yale.edu`

## Abstract

We provide the first generalization error analysis for black-box learning through derivative-free optimization. Under the assumption of a Lipschitz and smooth unknown loss, we consider the *Zeroth-order Stochastic Search* (ZoSS) algorithm, that updates a $d$-dimensional model by replacing stochastic gradient directions with stochastic differences of $K + 1$ perturbed loss evaluations per dataset (example) query. For both unbounded and bounded possibly nonconvex losses, we present the first generalization bounds for the ZoSS algorithm. These bounds coincide with those for SGD, and they are independent of $d$, $K$ and the batch size $m$, under appropriate choices of a slightly decreased learning rate. For bounded nonconvex losses and a batch size $m = 1$, we additionally show that both generalization error and learning rate are independent of $d$ and $K$, and remain essentially the same as for the SGD, even for two function evaluations. Our results extensively extend and consistently recover established results for SGD in prior work, on both generalization bounds and corresponding learning rates. If additionally $m = n$, where $n$ is the dataset size, we recover generalization guarantees for full-batch GD as well.

## 1 Introduction

Learning methods often rely on empirical risk minimization objectives that highly depend on a limited training data-set. Known gradient-based approaches such as SGD train and generalize effectively in reasonable time [1]. In contrast, emerging applications such as convex bandits [2–4], black-box learning [5], federated learning [6], reinforcement learning [7, 8], learning linear quadratic regulators [9, 10], and hyper-parameter tuning [11] stand in need of *gradient-free learning algorithms* [11–14] due to an unknown loss/model or impossible gradient evaluation.

Given two or more function evaluations, zeroth-order algorithms (see, e.g., [14, 15]) aim to estimate the true gradient for evaluating and updating model parameters (say, of dimension $d$). In particular, Zeroth-order Stochastic Search (ZoSS) [13, Corollary 2], [16, Algorithm 1] uses $K + 1$ function evaluations ($K \geq 1$), while deterministic zeroth-order approaches [5, Section 3.3] require at least $K \geq d + 1$ queries. The optimization error of the ZoSS algorithm is optimal as shown in prior work for convex problems [13], and suffers at most a factor of $\sqrt{d/K}$ in the convergence rate as compared with SGD. In addition to the optimization error, the importance of generalization error raises the question of how well zeroth-order algorithms generalize to unseen examples. In this paper, we show that the generalization error of ZoSS essentially coincides with that of SGD, under the choice of a slightly decreased learning rate. Assuming a Lipschitz and smooth loss function, we

establish generalization guarantees for ZoSS, by extending stability-based analysis for SGD [1], to the gradient-free setting. In particular, we rely on the celebrated result that uniform algorithmic stability implies generalization [1, 17, 18].

Early works [17, 19–22] first introduced the notion of stability, and the connection between (uniform) stability and generalization. Recently, alternative notions of stability and generalization gain attention such as locally elastic stability [23], VC-dimension/flatness measures [24], distributional stability [25–27], information theoretic bounds [16, 28–33] mainly based on assuming a sub-Gaussian loss, as well as connections between differential privacy and generalization [34–37].

In close relation to our paper, Hardt et al. [1] first showed uniform stability final-iterate bounds for vanilla SGD. More recent works develop alternative generalization error bounds based on high probability guarantees [38–41] and data-dependent variants [42], or under different assumptions than those of prior works such as as strongly quasi-convex [43], non-smooth convex [44–47], and pairwise losses [48, 49]. In the nonconvex case, [50] provide bounds that involve on-average variance of the stochastic gradients. Generalization performance of other algorithmic variants lately gain further attention, including SGD with early momentum [51], randomized coordinate descent [52], look-ahead approaches [53], noise injection methods [54], and stochastic gradient Langevin dynamics [55–62].

Recently, stability and generalization of full-bath GD has also been studied; see, e.g., [63–67]. In particular, Charles and Papailiopoulos. [64] showed instability of GD for nonconvex losses. Still, such instability does not imply a lower bound on the generalization error of GD (in expectation). In fact, Hoffer et al. [63] showed empirically that the generalization of GD is not affected by the batch-size, and for large enough number of iterations GD generalizes comparably to SGD. Our analysis agree with the empirical results of Hoffer et al. [63], as we show that (for smooth losses) the generalization of ZoSS (and thus of SGD) is independent of the batch size.

**Notation.** We denote the training data-set $S$ of size $n$ as $\{z_i\}_{i=1}^n$, where $z_i$ are i.i.d. observations of a random variable $Z$ with unknown distribution $\mathcal{D}$. The parameters of the model are vectors of dimension $d$, denoted by $W \in \mathbb{R}^d$, and $W_t$ is the output at time $t$ of a (randomized) algorithm $A_S$. The (combined) loss function $f(\cdot, z) : \mathbb{R}^d \to \mathbb{R}^+$ is uniformly Lipschitz and smooth for all $z \in \mathcal{Z}$. We denote the Lipschitz constant as $L$ and the smoothness parameter by $\beta$. The number of function (i.e., loss) evaluations (required at each iteration of the ZoSS algorithm) is represented by $K + 1 \in \mathbb{N}$. We denote by $\Delta f$ the smoothed approximation of the loss gradient, associated with parameter $\mu$. The parameter $\Gamma_K^d \triangleq \sqrt{(3d-1)/K + 1}$ prominently appears in our results. We denote the gradient of the loss function with respect to model parameters $W$, by $\nabla f(W, z) \equiv \nabla_w f(w, z)|_{w=W}$. We denote the mini batch at $t$ by $J_t$, and $m \triangleq |J_t|$.

## 1.1 Contributions

Under the assumption of Lipschitz and smooth loss functions, we provide generalization guarantees for black-box learning, extending the analysis of prior work by Hardt et al. [1] to the gradient free setting. In particular, we establish uniform stability and generalization error bounds for the final iterate of the ZoSS algorithm; see Table 1 for a summary of the results. In more detail, the contributions of this work are as follows:

- For unbounded *and* bounded losses, we show generalization error bounds identical to SGD, with a slightly decreased learning rate. Specifically, the generalization error bounds are independent of the dimension $d$, the number of evaluations $K$ and the batch-size $m$. Further, a large enough number of evaluations ($K$) provide fast generalization even in the high dimensional regime.
- For bounded nonconvex losses and single (example) query updates ($m = 1$), we show that both the ZoSS generalization error and learning rate are independent of $d$ and $K$, similar to that of SGD [1, Theorem 3.8]. This property guarantees efficient generalization even with two function evaluations.
- In the full information regime (i.e., when the number of function evaluations $K$ grow to $\infty$), the ZoSS generalization bounds also provide guarantees for SGD by recovering the results in prior work [1]. Further, we derive novel SGD bounds for unbounded nonconvex losses, as well as mini-batch SGD for any batch size. Our results subsume generalization guarantees for full-batch ZoSS and GD algorithms.

| Generalization Error Bounds: ZoSS vs SGD | | | | |
|---|---|---|---|---|
| Algorithm | Bound | NC | UB | MB |
| ZoSS (**this work**) $\alpha_t \le C/(t\Gamma_K^d)$ | $\dfrac{1+(C\beta)^{-1}}{n}\left((2+c)CL^2\right)^{\frac{1}{C\beta+1}}(eT)^{\frac{C\beta}{C\beta+1}}$ | ✓ | ✗ | ✗ |
| SGD, $\alpha_t \le C/t$ Hardt et al. [1] | $\dfrac{1+(C\beta)^{-1}}{n}\left(2CL^2\right)^{\frac{1}{C\beta+1}}(eT)^{\frac{C\beta}{C\beta+1}}$ | ✓ | ✗ | ✗ |
| ZoSS (**this work**) $\alpha_t \le C/t$ | $\dfrac{3e\left(1+(C\beta)^{-1}\right)^2}{2n}\left(1+(2+c)CL^2\right)T$ 
 (*independent of both $d$ and $K$*) | ✓ | ✗ | ✗ |
| ZoSS (**this work**) $\alpha_t \le \dfrac{\log\left(1+\frac{C\beta}{\Gamma_K^d}(\Gamma_K^d-1)\right)}{T\beta\sqrt{(3d-1)/K}}$ | $\dfrac{(2+c)CL^2}{n}$ | ✗ | ✓ | ✓ |
| SGD, $\alpha_t \le C/T$ Hardt et al. [1] | $\dfrac{2CL^2}{n}$ | ✗ | ✓ | ✓ |
| ZoSS (**this work**) $\alpha_t \le C/(T\Gamma_K^d)$ | $\dfrac{(2+c)L^2(e^{C\beta}-1)}{n\beta}$ | ✓ | ✓ | ✓ |
| ZoSS (**this work**) $\alpha_t \le \dfrac{\log(1+C\beta)}{T\beta\Gamma_K^d}$ | $\dfrac{(2+c)CL^2}{n}$ 
 (proper choice of $C$ in previous bound) | ✓ | ✓ | ✓ |
| ZoSS (**this work**) $\alpha_t \le C/(t\Gamma_K^d)$ | $\dfrac{(2+c)L^2(eT)^{C\beta}}{n}\min\{C+\beta^{-1},C\log(T)\}$ | ✓ | ✓ | ✓ |

Table 1: A list of the generalization error bounds developed herein for ZoSS (Eq. 6) in comparison with SGD, with $\mu \le cL\Gamma_K^d/n\beta(3+d)^{3/2}$, for $c > 0$. In the table, "NC" and "UB" stand for "nonconvex" and "unbounded", respectively. "MB" corresponds to the mini-batch algorithm and for any batch size. Also, $\alpha_t$ denotes the stepsize of ZoSS/SGD, and $T$ the total number of iterations.

## 2  Problem Statement

Given a data $S \triangleq \{z_i\}_{i=1}^n$ of i.i.d samples $z_i$ from an unknown distribution $\mathcal{D}$, our goal is to find the parameters $w^*$ of a learning model such that $w^* \in \arg\min_w R(w)$, where $R(w) \triangleq \mathbb{E}_{Z\sim\mathcal{D}}[f(w,Z)]$. Since the distribution $\mathcal{D}$ is not known, we consider the empirical risk

$$R_S(w) \triangleq \frac{1}{n}\sum_{i=1}^n f(w,z_i), \tag{1}$$

and the corresponding empirical risk minimization (ERM) problem to find $w_s^* \in \arg\min_w R_S(w)$. For a (randomized) algorithm $A_S$ with input $S$ and output $W = A(S)$, the excess risk $\epsilon_{\text{excess}}$ is bounded by the sum of the generalization error $\epsilon_{\text{gen}}$ and the optimization error $\epsilon_{\text{opt}}$,

$$\epsilon_{\text{excess}} \triangleq \mathbb{E}_{S,A}[R(W)] - R(w^*) = \underbrace{\mathbb{E}_{S,A}[R(W) - R_S(W)]}_{\epsilon_{\text{gen}}} + \underbrace{(\mathbb{E}_{S,A}[R_S(W)] - R(w^*))}_{\epsilon_{\text{opt}}}. \tag{2}$$

To analyze and control $\epsilon_{\text{gen}}$, we prove uniform stability bounds which imply generalization [1, Theorem 2.2]. Specifically, if for all i.i.d. sequences $S, S' \in \mathcal{Z}^n$ that differ in one entry, we have $\sup_z \mathbb{E}_A[f(A(S),z) - f(A(S'),z)] \le \epsilon_{\text{stab}}$, for some $\epsilon_{\text{stab}} > 0$, then $\epsilon_{\text{gen}} \le \epsilon_{\text{stab}}$. Because the loss is $L$-Lipschitz, $\epsilon_{\text{stab}}$ may then be chosen as $L\sup_{S,S'}\mathbb{E}_A\|A(S) - A(S')\|$.

Our primary goal in this work is to develop uniform stability bounds for a *gradient-free* algorithm $A_S$ of the form $w_{t+1} = w_t - \alpha_t\Delta f_{w_t,z}$, where $\Delta f_{w_t,z}$ only depends on loss function evaluations. To achieve this without introducing unnecessary assumptions, we consider a novel algorithmic stability error decomposition approach. In fact, the stability error introduced at time $t$ by $A_S$ breaks down into the stability error of SGD and an approximation error due to missing gradient information. Let $G_t(\cdot)$

and $G'_t(\cdot)$ be the following SGD update rules

$$G_t(w) \triangleq w - \alpha_t \nabla f(w, z_{i_t}), \quad G'_t(w) \triangleq w - \alpha_t \nabla f(w, z'_{i_t}), \tag{3}$$

under inputs $S, S'$ respectively, and let $i_t \in \{1, 2, \ldots, n\}$ be a random index chosen uniformly and independently by the random selection rule of the algorithm, for all $t \le T$. Similarly we use the notation $\tilde{G}(\cdot)$ and $\tilde{G}'(\cdot)$ to denote the iteration mappings of $A_S$, i.e.,

$$\tilde{G}_t(w) \triangleq w - \alpha_t \Delta f_{w,z_{i_t}}, \quad \tilde{G}'_t(w) \triangleq w - \alpha_t \Delta f_{w,z'_{i_t}}. \tag{4}$$

Then, as we also discuss later on (Lemma 1), the iterate stability error $\tilde{G}_t(w) - \tilde{G}'_t(w')$ of $A_S$, for any $w, w' \in \mathbb{R}^d$ and for all at $t \le T$, may be decomposed as

$$\tilde{G}_t(w) - \tilde{G}'_t(w') \propto \underbrace{G_t(w) - G'_t(w')}_{\epsilon_{\text{GBstab}}} + \underbrace{\left[\nabla f(w, z_{i_t}) - \Delta f_{w,z_{i_t}}\right] + \left[\nabla f(w', z'_{i_t}) - \Delta f_{w',z'_{i_t}}\right]}_{\epsilon_{\text{est}}}, \tag{5}$$

where $\epsilon_{\text{GBstab}}$ denotes the gradient-based stability error (associated with SGD), and $\epsilon_{\text{est}}$ denotes the gradient approximation error. We now proceed by formally introducing ZoSS.

# 3 Zeroth-Order Stochastic Search (ZoSS)

As a *gradient-free* alternative of the classical SGD algorithm, we consider the ZoSS scheme, with iterates generated according to the following (single-example update) rule

$$W_{t+1} = W_t - \alpha_t \frac{1}{K} \sum_{k=1}^{K} \frac{f(W_t + \mu U^t_k, z_{i_t}) - f(W_t, z_{i_t})}{\mu} U^t_k, \quad U^t_k \sim \mathcal{N}(0, I_d), \quad \mu \in \mathbb{R}^+, \tag{6}$$

where $\alpha_t \ge 0$ is the corresponding learning rate (for the mini-batch update rule we refer the reader to Section 5). At every iteration $t$, ZoSS generates $K$ i.i.d. standard normal random vectors $U^t_k, k = 1, \ldots, K$, and obtains $K + 1$ loss evaluations on perturbed model inputs. Then ZoSS evaluates a smoothed approximation of the gradient for some $\mu > 0$. In light of the discussion in Section 2, we define the ZoSS smoothed gradient step at time $t$ as

$$\Delta f^{K,\mu}_{w,z_{i_t}} \equiv \Delta f^{K,\mu,\mathbf{U}^t}_{w,z_{i_t}} \triangleq \frac{1}{K} \sum_{k=1}^{K} \frac{f(w + \mu U^t_k, z_{i_t}) - f(w, z_{i_t})}{\mu} U^t_k. \tag{7}$$

## 3.1 ZoSS Stability Error Decomposition

To show stability bounds for ZoSS, we decompose its error into two parts through the stability error decomposition discussed in Section 2. Under the ZoSS update rule, Eq. (5) holds by considering the directions $\Delta f_{w,z_{i_t}}$ and $\Delta f_{w',z'_{i_t}}$ according to ZoSS smoothed approximations (7). Then for any $w, w' \in \mathbb{R}^d$, the iterate stability error $\tilde{G}_t(w) - \tilde{G}'_t(w')$ of ZoSS at $t$, breaks down into the gradient based error $\epsilon_{\text{GBstab}}$ and approximation error $\epsilon_{\text{est}}$.

The error term $\epsilon_{\text{GBstab}}$ expresses the stability error of the gradient based mappings [1, Lemma 2.4] and inherits properties related to the SGD update rule. The error $\epsilon_{\text{est}}$ captures the approximation error of the ZoSS smoothed approximation and depends on $K$ and $\mu$. The consistency of the smoothed approximation with respect to SGD follows from $\lim_{K \uparrow \infty, \mu \downarrow 0} \Delta f^{K,\mu}_{w,z} = \nabla f(w, z)$ for all $w \in \mathbb{R}$ and $z \in \mathcal{Z}$. Further, the stability error is also consistent since $\lim_{K \uparrow \infty, \mu \downarrow 0} |\epsilon_{\text{est}}| = 0$. Later on, we use the ZoSS error decomposition in Eq. (5) together with a variance reduction lemma (Lemma 10), to derive exact expressions on the iterate stability error $\tilde{G}_t(w) - \tilde{G}'_t(w')$ for fixed $K$ and $\mu > 0$ (see Lemma 1). Although in this paper we derive stability bounds and bounds on the $\epsilon_{\text{gen}}$, the excess risk $\epsilon_{\text{excess}}$ depends on both errors $\epsilon_{\text{gen}}$ and $\epsilon_{\text{opt}}$. In the following section, we briefly discuss known results on the $\epsilon_{\text{opt}}$ of zeroth-order methods, including convex and nonconvex losses.

## 3.2 Optimization Error in Zeroth-Order Stochastic Approximation

Convergence rates of the ZoSS optimization error and related zeroth-order variants have been extensively studied in prior works; see e.g., [14, 15, 68]. For the convex loss setting, when $K + 1$

function evaluations are available and no other information regarding the loss is given, the ZoSS algorithm achieves optimal rates with respect to the optimization error $\epsilon_{\text{opt}}$. Specifically, under the assumption of a closed and convex loss, Duchi et al. [13] provided a lower bound for the minimax convergence rate and showed that $\epsilon_{\text{opt}} = \Omega(\sqrt{d/K})$, for any algorithm that approximates the gradient given $K + 1$ evaluations. In the nonconvex setting Ghadimi et al. [69, 70] established sample complexity guarantees for the zeroth-order approach to reach an approximate stationary point.

## 4   Main Results

For our analysis, we introduce the same assumptions on the loss function (Lipschitz and smooth) as appears in prior work [1]. Additionally, we exploit the $\eta$-expansive and $\sigma$-bounded properties of the SGD mappings $G_t(\cdot)$ and $G_t'(\cdot)$ in Eq. (3).[1] The mappings $G_t(\cdot)$ and $G_t'(\cdot)$ are introduced for analysis purposes due to the stability error decomposition given in Eq. (5) and no further assumptions or properties are required for the zeroth-order update rules $\tilde{G}_t(\cdot)$ and $\tilde{G}_t'(\cdot)$ given in Eq. (4). The $\eta$-expansivity of $G_t(\cdot)$ holds for $\eta = 1 + \beta\alpha_t$ if the loss is nonconvex, and $\eta = 1$ if the loss is convex and $\alpha_t \le 2/\beta$ [1, Lemma 3.6]. Note that $G_t(\cdot)$ is always $\sigma$-bounded ($\sigma = L\alpha_t$) [1, Lemma 3.3.].

### 4.1   Stability Analysis

We derive generalization error bounds through uniform stability. To study the stability of ZoSS, we apply a variance reduction lemma that we provide in Appendix A. Exploiting the variance reduction lemma, we show a growth recursion lemma for the iterates of the ZoSS.

**Lemma 1 (ZoSS Growth Recursion)** *Consider the sequences of updates $\{\tilde{G}_t\}_{t=1}^T$ and $\{\tilde{G}_t'\}_{t=1}^T$. Let $w_0 = w_0'$ be the starting point, $w_{t+1} = \tilde{G}_t(w_t)$ and $w_{t+1}' = \tilde{G}_t'(w_t')$ for any $t \in \{1, \dots, T\}$. Then for any $w_t, w_t' \in \mathbb{R}^d$ and $t \ge 0$ the following recursion holds*

$$\mathbb{E}[\|\tilde{G}_t(w_t) - \tilde{G}_t'(w_t')\|] \le \begin{cases} \left(\eta + \alpha_t \sqrt{\frac{3d-1}{K}}\beta\right)\|w_t - w_t'\| + \mu\beta\alpha_t(3+d)^{3/2}, & \text{if } \tilde{G}_t(\cdot) = \tilde{G}_t'(\cdot), \\ \|w_t - w_t'\| + 2\alpha_t L\Gamma_K^d + \mu\beta\alpha_t(3+d)^{3/2}, & \text{if } \tilde{G}_t(\cdot) \ne \tilde{G}_t'(\cdot). \end{cases}$$

The growth recursion of ZoSS characterizes the stability error that it is introduced by the ZoSS update and according to the outcome of the randomized selection rule at each iteration. Lemma 1 extends growth recursion results for SGD in prior work [1, Lemma 2.5] to the setting of the ZoSS algorithm. If $K \to \infty$ and $\mu \to 0$ (while the rest of the parameters are fixed), then $\Gamma_K^d \to 1$, and the statement recovers that of the SGD [1, Lemma 2.5].

**Proof of Lemma 1.**   Let $S$ and $S'$ be two samples of size $n$ differing in only a single example, and let $\tilde{G}_t(\cdot), \tilde{G}_t'(\cdot)$ be the update rules of the ZoSS for each of the sequences $S, S'$ respectively. First under the event $\mathcal{E}_t \triangleq \{\tilde{G}_t(\cdot) \equiv \tilde{G}_t'(\cdot)\}$ (see Eq. (4)), by applying the Taylor expansion there exist vectors $W_{k,t}^*$ and $W_{k,t}^\dagger$ with $j^{\text{th}}$ coordinates in the intervals $\left(w_t^{(j)}, w_t^{(j)} + \mu U_{k,t}^{(j)}\right) \cup \left(w_t^{(j)} + \mu U_{k,t}^{(j)}, w_t^{(j)}\right)$ and $\left(w_t'^{(j)}, w_t'^{(j)} + \mu U_{k,t}^{(j)}\right) \cup \left(w_t'^{(j)} + \mu U_{k,t}^{(j)}, w_t'^{(j)}\right)$, respectively, such we find that for any $w_t, w_t' \in \mathbb{R}^d$ it is true that

$$\tilde{G}_t(w_t) - \tilde{G}_t'(w_t') = \tilde{G}_t(w_t) - \tilde{G}_t(w_t')$$

$$= w_t - w_t' - \frac{\alpha_t}{K}\sum_{k=1}^K \langle \nabla f(w_t, z_{i_t}) - \nabla f(w_t', z_{i_t}), U_k^t\rangle U_k^t \tag{8}$$

$$- \frac{\alpha_t}{K}\sum_{k=1}^K \left(\frac{\mu}{2}U_k^{\text{T}}\nabla_w^2 f(w, z_{i_t})|_{w=W_{k,t}^*} U_k^t\right) U_k^t + \frac{\alpha_t}{K}\sum_{k=1}^K \left(\frac{\mu}{2}U_k^{\text{T}}\nabla_w^2 f(w, z_{i_t})|_{w=W_{k,t}^\dagger} U_k^t\right) U_k^t$$

$$= \underbrace{w_t - \alpha_t\nabla f(w_t, z_{i_t})}_{G(w_t)} - \underbrace{(w_t' - \alpha_t\nabla f(w_t', z_{i_t}))}_{G'(w_t')\equiv G(w_t')}$$

---

[1] [1, Definition 2.3]: An update rule $G(\cdot)$ is $\eta$-expansive if $\|G(w) - G(w')\| \le \eta\|w - w'\|$ for all $w, w' \in \mathbb{R}^d$. If $\|w - G(w)\| \le \sigma$ then it is $\sigma$-bounded.

$$- \frac{\alpha_t}{K} \sum_{k=1}^{K} \left( \frac{\mu}{2} U_k^{\mathrm{T}} \nabla_w^2 f(w, z_{i_t})|_{w=W_{k,t}^*} U_k^t \right) U_k^t + \frac{\alpha_t}{K} \sum_{k=1}^{K} \left( \frac{\mu}{2} U_k^{\mathrm{T}} \nabla_w^2 f(w, z_{i_t})|_{w=W_{k,t}^{\dagger}} U_k^t \right) U_k^t$$

$$- \alpha_t \left( \frac{1}{K} \sum_{k=1}^{K} \langle \nabla f(w_t, z_{i_t}) - \nabla f(w_t', z_{i_t}), U_k^t \rangle U_k^t - (\nabla f(w_t, z_{i_t}) - \nabla f(w_t', z_{i_t})) \right). \tag{9}$$

We find (9) by adding and subtracting $\alpha_t \nabla f(w_t, z_{i_t})$ and $\alpha_t \nabla f(w_t', z_{i_t})$ in Eq. (8). Recall that $U_k^t$ are independent for all $k \leq K$, $t \leq T$ and that the mappings $G(\cdot)$ and $G'(\cdot)$ defined in Eq. (9), are $\eta$-expansive. The last display and the triangle inequality give

$$\mathbb{E}[\|\tilde{G}_t(w_t) - \tilde{G}_t(w_t')\|]$$

$$\leq \|G(w_t) - G(w_t')\| + \frac{2\alpha_t}{K} \sum_{k=1}^{K} \frac{\mu\beta}{2} \mathbb{E}\left[\|U_k^t\|^3\right] + \alpha_t \sqrt{\frac{3d-1}{K}} \mathbb{E}[\|\nabla f(w_t, z_{i_t}) - \nabla f(w_t', z_{i_t})\|] \tag{10}$$

$$\leq \eta\|w_t - w_t'\| + \frac{2\alpha_t}{K} \sum_{k=1}^{K} \frac{\mu\beta}{2} \mathbb{E}\left[\|U_k^t\|^3\right] + \alpha_t \sqrt{\frac{3d-1}{K}} \beta\|w_t - w_t'\| \tag{11}$$

$$\leq \left( \eta + \alpha_t \sqrt{\frac{3d-1}{K}} \beta \right) \|w_t - w_t'\| + \mu\beta\alpha_t(3+d)^{3/2}, \tag{12}$$

where (10) follows from (9) and Lemma 10, and for (11) we applied the $\eta$-expansive property of $G(\cdot)$ (see [1, Lemma 2.4 and Lemma 3.6]) and the $\beta$-smoothness of the loss function.[2] Finally (12) holds since the random vectors $U_k^t \sim \mathcal{N}(0, I_d)$ are identically distributed for all $k \in \{1, 2, \ldots, K\}$ and $\mathbb{E}\|U_k^t\|^3 \leq (3+d)^{3/2}$. Eq. (12) gives the first part of the recursion.

Similar to (9), under the event $\mathcal{E}_t^c \triangleq \{\tilde{G}_t(\cdot) \neq \tilde{G}_t'(\cdot)\}$, we find

$$\tilde{G}_t(w_t) - \tilde{G}_t'(w_t')$$

$$= \underbrace{w_t - \alpha_t \nabla f(w_t, z_{i_t})}_{G(w_t)} - \underbrace{(w_t' - \alpha_t \nabla f(w_t', z_{i_t}'))}_{G'(w_t')}$$

$$- \frac{\alpha_t}{K} \sum_{k=1}^{K} \left( \frac{\mu}{2} U_k^{\mathrm{T}} \nabla_w^2 f(w, z_{i_t})|_{w=\tilde{W}_{k,t}^*} U_k^t \right) U_k^t + \frac{\alpha_t}{K} \sum_{k=1}^{K} \left( \frac{\mu}{2} U_k^{\mathrm{T}} \nabla_w^2 f(w, z_{i_t}')|_{w=\tilde{W}_{k,t}^{\dagger}} U_k^t \right) U_k^t$$

$$- \alpha_t \left( \frac{1}{K} \sum_{k=1}^{K} \langle \nabla f(w_t, z_{i_t}) - \nabla f(w_t', z_{i_t}'), U_k^t \rangle U_k^t - (\nabla f(w_t, z_{i_t}) - \nabla f(w_t', z_{i_t}')) \right). \tag{13}$$

By using the last display, triangle inequality, Lemma 10 and $\beta$-smoothness, we find

$$\mathbb{E}[\|\tilde{G}_t(w_t) - \tilde{G}_t(w_t')\|]$$

$$\leq \|G(w_t) - G'(w_t')\| + \frac{2\alpha_t}{K} \sum_{k=1}^{K} \frac{\mu\beta}{2} \mathbb{E}[\|U_k^t\|^3] + \alpha_t \sqrt{\frac{3d-1}{K}} \mathbb{E}[\|\nabla f(w_t, z_{i_t}) - \nabla f(w_t', z_{i_t}')\|]$$

$$\leq \min\{\eta, 1\}\delta_t + 2\sigma_t + \frac{2\alpha_t}{K} \sum_{k=1}^{K} \frac{\mu\beta}{2} \mathbb{E}[\|U_k^t\|^3] + 2L\alpha_t \sqrt{\frac{3d-1}{K}} \tag{14}$$

$$\leq \delta_t + 2\alpha_t L\Gamma_K^d + \mu\beta\alpha_t(3+d)^{3/2}, \tag{15}$$

where (14) follows from the triangle inequality and $L-$Lipschitz condition, while the upper bound on $\|G(w_t) - G'(w_t')\|$ comes from [1, Lemma 2.4]. Finally, (15) holds since $\eta \geq 1$ for both convex and nonconvex losses, $\sigma_t = L\alpha_t$ and $\mathbb{E}\|U_k^t\|^3 \leq (3+d)^{3/2}$ for all $k \in \{1, \ldots, K\}$. This shows the second part of recursion. $\square$

For sake of brevity, let $\mathcal{I}$ be an adapted stopping time that corresponds to the first iteration index that the single distinct instance of the two data-sets $S, S'$ is sampled by ZoSS. For any $t_0 \in \{0, 1, \ldots, n\}$ we define the event $\mathcal{E}_{\delta_{t_0}} \triangleq \{\mathcal{I} > t_0\} \equiv \{\delta_{t_0} = 0\}$. The next result provides the stability bound.

---

[2] For all $z \in \mathcal{Z}$ and $W \in \mathbb{R}^d$ it is true that $\|\nabla_w^2 f(w, z)|_{w=W}\| \leq \beta$.

**Lemma 2 (ZoSS Stability | Nonconvex Loss)** *Assume that the loss function $f(\cdot, z)$ is L-Lipschitz and $\beta$-smooth for all $z \in \mathcal{Z}$. Consider the ZoSS algorithm (6) with final-iterate estimates $W_T$ and $W_T'$, corresponding to the data-sets $S, S'$, respectively (that differ in exactly one entry). Then the discrepancy $\delta_T \triangleq \|W_T - W_T'\|$, under the event $\mathcal{E}_{\delta_{t_0}}$, satisfies the inequality*

$$\mathbb{E}[\delta_T | \mathcal{E}_{\delta_{t_0}}] \leq \left( \frac{2L}{n} \Gamma_K^d + \mu\beta(3+d)^{3/2} \right) \sum_{t=t_0+1}^{T} \alpha_t \prod_{j=t+1}^{T} \left( 1 + \beta\alpha_j \Gamma_K^d \left( 1 - \frac{1}{n} \right) \right). \quad (16)$$

The corresponding bound of Lemma 2 for convex losses is slightly tighter than the bound in (16). Since the two bounds differ only by a constant, the consequent results of Lemma 2 are essentially identical for convex losses as well. We provide the equivalent version of Lemma 2 for convex losses in Appendix B.

**Proof of Lemma 2.** Consider the events $\mathcal{E}_t \triangleq \{\tilde{G}_t(\cdot) \equiv \tilde{G}_t'(\cdot)\}$ and $\mathcal{E}_t^c \triangleq \{\tilde{G}_t(\cdot) \neq \tilde{G}_t'(\cdot)\}$ (see Eq. (4)). Recall that $\mathbb{P}(\mathcal{E}_t) = 1 - 1/n$ and $\mathbb{P}(\mathcal{E}_t^c) = 1/n$ for all $t \leq T$. For any $t_0 \geq 0$, a direct application of Lemma 1 gives

$$\mathbb{E}[\delta_{t+1} | \mathcal{E}_{\delta_{t_0}}] = \mathbb{P}(\mathcal{E}_t) \mathbb{E}[\delta_{t+1} | \mathcal{E}_t, \mathcal{E}_{\delta_{t_0}}] + \mathbb{P}(\mathcal{E}_t^c) \mathbb{E}[\delta_{t+1} | \mathcal{E}_t^c, \mathcal{E}_{\delta_{t_0}}]$$

$$= \left( 1 - \frac{1}{n} \right) \mathbb{E}[\delta_{t+1} | \mathcal{E}_t, \mathcal{E}_{\delta_{t_0}}] + \frac{1}{n} \mathbb{E}[\delta_{t+1} | \mathcal{E}_t^c, \mathcal{E}_{\delta_{t_0}}]$$

$$\leq \left( \eta + \alpha_t \beta \sqrt{\frac{3d-1}{K}} + \frac{1}{n} \left( 1 - \eta - \alpha_t \beta \sqrt{\frac{3d-1}{K}} \right) \right) \mathbb{E}[\delta_t | \mathcal{E}_{\delta_{t_0}}]$$

$$+ \frac{2\alpha_t L}{n} \Gamma_K^d + \mu\beta\alpha_t(3+d)^{3/2}. \quad (17)$$

With $R_t \triangleq (\eta + \alpha_t \beta(\Gamma_K^d - 1) + (1 - \eta - \alpha_t\beta(\Gamma_K^d - 1))/n)$ solving the recursion in (17) gives

$$\mathbb{E}[\delta_T | \mathcal{E}_{\delta_{t_0}}] \leq \left( \frac{2L}{n} \Gamma_K^d + \mu\beta(3+d)^{3/2} \right) \sum_{t=t_0+1}^{T} \alpha_t \prod_{j=t+1}^{T} R_j. \quad (18)$$

We consider the last inequality for nonconvex loss functions with $\eta = 1 + \beta\alpha_t$ and convex loss functions with $\eta = 1$ to derive Lemma 2 and Lemma 11 respectively (Appendix B). $\qquad\square$

## 4.2 Generalization Error Bounds

For the first generalization error bound, we evaluate the right part of the inequality (16) for decreasing step size and bounded nonconvex loss. Then the Lipschitz condition provides a uniform stability condition for the loss and yields the next theorem.

**Theorem 3 (Nonconvex Bounded Loss | Decreasing Stepsize)** *Assume that the loss $f(\cdot, z) \in [0, 1]$ is L-Lipschitz and $\beta$-smooth for all $z \in \mathcal{Z}$. Consider the ZoSS update rule (6) with $T$ the total number of iterates, $\alpha_t \leq C/t\Gamma_K^d$ for some (fixed) $C > 0$ and for all $t \leq T$, and fixed $\mu \leq cL\Gamma_K^d/n\beta(3+d)^{3/2}$ for some $c > 0$. Then the generalization error of ZoSS is bounded by*

$$|\epsilon_{\text{gen}}| \leq \frac{\left( (2+c)CL^2 \right)^{\frac{1}{C\beta+1}} (eT)^{\frac{C\beta}{C\beta+1}}}{n} \max\left\{ 1, 1 + (C\beta)^{-1} - \frac{e^{\beta C}}{\beta C^{\frac{1}{C\beta+1}}} \left( \frac{(2+c)L^2}{eT} \right)^{\frac{C\beta}{C\beta+1}} \right\} \quad (19)$$

$$\leq \frac{\left( 1 + (C\beta)^{-1} \right) \left( (2+c)CL^2 \right)^{\frac{1}{C\beta+1}}}{n} (eT)^{\frac{C\beta}{C\beta+1}}. \quad (20)$$

Inequality (19), as a tighter version of (20), provides a meaningful bound in marginal cases, i.e.,

$$\lim_{\beta \downarrow 0} \mathbb{E}\left[ |f(W_T, z) - f(W_T', z)| \right] \leq \frac{(2+c)CL^2}{n} \max\left\{ \log\left( \frac{eT}{(2+c)CL^2} \right), 1 \right\}. \quad (21)$$

By neglecting the negative term in (19) we find (20), that is the ZoSS equivalent of SGD [1, Theorem 3.8]. When $K \to \infty$ and $c \to 0$, then $\Gamma_K^d \to 1$, and the inequalities (19), (20) reduce to a

generalization bound for SGD. Inequality (20) matches that of [1, Theorem 3.8], and (19) provides a tighter generalization bound for SGD as well. We show Theorem 3 in Appendix A.

Next, we provide a bound on the generalization error for nonconvex losses that comes directly from Theorem 3. In contrast to Theorem 3, the next result provides learning rate and a generalization error bounds, both of which are independent of the dimension and the number of function evaluations.

**Corollary 4** *Assume that the loss function $f(\cdot, z) \in [0, 1]$ is L-Lipschitz and $\beta$-smooth for all $z \in \mathcal{Z}$. Consider the ZoSS update rule (6) with $\mu \leq cL\Gamma_K^d/(n\beta(3 + d)^{3/2})$, $T$ the total number of iterates, and $\alpha_t \leq C/t$ for some (fixed) $C > 0$ and for all $t \leq T$. Then the generalization error of ZoSS is bounded by*

$$|\epsilon_{\text{gen}}| \leq \left(1 + (\beta C)^{-1}\right)^2 \left(1 + (2 + c)CL^2\right) \frac{3Te}{2n}. \tag{22}$$

As a consequence, even in the high dimensional regime $d \to \infty$, two function evaluations (i.e., $K = 1$) are sufficient for the ZoSS to achieve $\epsilon_{\text{gen}} = \mathcal{O}(T/n)$, with the learning rate being no smaller than that of SGD. We continue by providing the proof of Theorem 3. For the proof of Corollary 4, see Appendix A.3.

In light of Theorem 3 and Corollary 4, we observe that the over-fitting phenomenon occurs in the gradient-free approach similarly to gradient-based algorithms. For general nonconvex (and convex) losses under standard step-size choices, the generalization error increases with respect to $T$. Further, the effect of $\beta$ affects both the stability (similarly to SGD in prior work) of the algorithm and the error approximation of the ZoSS. If $\beta$ is large, then the expected approximation error (due to limited function evaluations) is also large [15] and the dependence on smoothness is unavoidable in black-box learning. In our results, this is expressed through the Growth Recursion of ZoSS (Lemma 2), that involves both the stability and approximation error per iteration. However, a smaller step-size ($\alpha_t = 1/2t\beta\Gamma_K^d$) mitigates the effect of $\beta$ on the bound. We refer the reader to Appendix E for a unified analysis of the excess risk, that captures the over-fitting and under-fitting trade-off.

Additionally, the number of iterations $T$ is considered to be fixed and known (as in prior works including on average and high probability results on generalization). This is reasonable and quite standard because given the theoretical results, we know beforehand the appropriate choices of $T$ that provide a good trade-off between generalization and optimization. A classical setting is that of a fixed step-size $\alpha_t = 1/T$ with $T = \sqrt{n}$, which provides the well known generalization error bound for SGD with order $\mathcal{O}(1/\sqrt{n})$, as appears in very recent and timely prior works [32, Section 3.1], [45].

In the unbounded loss case, we apply Lemma 2 by setting $t_0 = 0$ (recall that $t_0$ is a free parameter, while the algorithm depends on the random variable $\mathcal{I}$). The next result provides a generalization error bound for the ZoSS algorithm with constant step size. In the first case of the theorem, we also consider the convex loss as a representative result, as we show the same bound holds for an appropriate choice of greater learning rate than the learning rate of the nonconvex case. The convex case for the rest of the results of this work can be similarly derived.

**Theorem 5 (Unbounded Loss | Constant Step Size)** *Assume that the loss $f(\cdot, z)$ is L-Lipschitz, $\beta$-smooth for all $z \in \mathcal{Z}$. Consider the ZoSS update rule (6) with $\mu \leq cL\Gamma_K^d/(n\beta(3 + d)^{3/2})$ for some $c > 0$. Let $T$ be the total number of iterates and for any $t \leq T$,*

• *if $f(\cdot, z)$ is convex for all $z \in \mathcal{Z}$ and $\alpha_t \leq \min\{\log\left(1 + C\beta(1 - 1/\Gamma_K^d)\right)/T\beta(\Gamma_K^d - 1), 2/\beta\}$, or if $f(\cdot, z)$ is nonconvex and $\alpha_t \leq \log\left(1 + C\beta\right)/T\beta\Gamma_K^d$, for $C > 0$ then*

$$|\epsilon_{\text{gen}}| \leq \frac{(2 + c)CL^2}{n}, \tag{23}$$

• *if $f(\cdot, z)$ is nonconvex and $\alpha_t \leq C/T\Gamma_K^d$, for some $C > 0$, then*

$$|\epsilon_{\text{gen}}| \leq \frac{L^2 (2 + c)\left(e^{C\beta} - 1\right)}{n\beta}. \tag{24}$$

For the proof of Theorem 5 see Appendix A.4. In the following, we present the generalization error of ZoSS for an unbounded loss with a decreasing step size. Recall that the results for unbounded nonconvex loss also hold for the case of a convex loss with similar bounds on the generalization error and learning rate (see the first case of Theorem 5).

**Theorem 6 (Unbounded Loss | Decreasing Step Size)** *Assume that the loss $f(\cdot, z)$ is L-Lipschitz, $\beta$-smooth for all $z \in \mathcal{Z}$. Consider ZoSS with update rule* (6), *$T$ the total number of iterates, $\alpha_t \leq C/t\Gamma_K^d$ for all $t \leq T$ and for some $C > 0$, and $\mu \leq cL\Gamma_K^d/(n\beta(3+d)^{3/2})$ for some $c > 0$. Then the generalization error of ZoSS is bounded by*

$$|\epsilon_{\text{gen}}| \leq \frac{(2+c)L^2(eT)^{C\beta}}{n} \min\left\{C + \beta^{-1}, C\log(eT)\right\}. \tag{25}$$

For the proof of Theorem 6 see Appendix A.5. Note that the constant $C$ is free and controls the learning rate. Furthermore, it quantifies the trade-off between the speed of training and the generalization of the algorithm. In the next section, we consider the ZoSS algorithm with a mini-batch of size $m$ for which we provide generalization error bounds. These results hold under the assumption of unbounded loss and for any batch size $m$ including the case $m = 1$.

## 5 Generalization of Mini-Batch ZoSS

For the *mini-batch* version of ZoSS, at each iteration $t$, the randomized selection rule (uniformly) samples a batch $J_t$ of size $m$ and evaluates the new direction of the update by averaging the smoothed approximation $\Delta f_{w,z}^{K,\mu}$ over the samples $z \in J_t$ as

$$\Delta f_{w,J_t}^{K,\mu} \equiv \Delta f_{w,J_t}^{K,\mu,\mathbf{U}^t} \triangleq \frac{1}{mK}\sum_{i=1}^{m}\sum_{k=1}^{K}\frac{f(w + \mu U_{k,i}^t, z_{J_{t,i}}) - f(w, z_{J_{t,i}})}{\mu}U_{k,i}^t, \tag{26}$$

where $U_{k,i}^t \sim \mathcal{N}(0, I_d)$ are i.i.d. (standard normal), and $\mu \in \mathbb{R}^+$. The update rule of the mini-batch ZoSS is $W_{t+1} = W_t - \alpha_t \Delta f_{W_t,J_t}^{K,\mu}$ for all $t \leq T$, and we define $\tilde{G}_{J_t}(w) \triangleq w - \alpha_t \Delta f_{w,J_t}^{K,\mu}$, $\tilde{G}'_{J'_t}(w) \triangleq w - \alpha_t \Delta f_{w,J'_t}^{K,\mu}$ for $J_t \subset S$ and $J'_t \subset S'$ respectively. Due to space limitation, we refer the reader to Appendix C for the detailed stability analysis of ZoSS with mini-batch. Specifically, we prove a growth recursion lemma for the mini-batch ZoSS updates (see Appendix C.1 for proof).

**Lemma 7 (Mini-Batch ZoSS Growth Recursion)** *Consider the sequences of updates $\{\tilde{G}_{J_t}\}_{t=1}^T$ and $\{\tilde{G}'_{J_t}\}_{t=1}^T$ and $\mu \leq cL\Gamma_K^d/(n\beta(3+d)^{3/2})$. Let $w_0 = w'_0$ be the starting point, $w_{t+1} = \tilde{G}_{J_t}(w_t)$ and $w'_{t+1} = \tilde{G}'_{J_t}(w'_t)$ for any $t \in \{1, \ldots, T\}$. Then for any $w_t, w'_t \in \mathbb{R}^d$ and $t \geq 0$ the following recursion holds*

$$\mathbb{E}[\|\tilde{G}_{J_t}(w_t) - \tilde{G}'_{J_t}(w'_t)\|] \leq \begin{cases} \left(1 + \beta\alpha_t\Gamma_K^d\right)\delta_t + \frac{cL\alpha_t}{n}\Gamma_K^d & \text{if } \tilde{G}_{J_t}(\cdot) = \tilde{G}'_{J_t}(\cdot) \\ \left(1 + \frac{m-1}{m}\beta\alpha_t\Gamma_K^d\right)\delta_t + \frac{2L\alpha_t}{m}\Gamma_K^d + \frac{cL\alpha_t}{n}\Gamma_K^d & \text{if } \tilde{G}_{J_t}(\cdot) \neq \tilde{G}'_{J_t}(\cdot). \end{cases}$$

Although the iterate stability error (at time $t$) in the growth recursion depends on the batch size $m$ under the event $\{\tilde{G}_{J_t}(\cdot) \neq \tilde{G}'_{J_t}(\cdot)\}$, the stability bound on the final iterates is independent of $m$, and coincides with the single example updates ($m = 1$, Lemma 2). Herein, we provide an informal statement of the result.

**Lemma 8 (Mini-Batch ZoSS Stability | Nonconvex Loss)** *Consider the mini-batch ZoSS with any batch size $m \leq n$, and iterates $W_{t+1} = W_t - \alpha_t \Delta f_{W_t,J_t}^{K,\mu}$, $W'_{t+1} = W'_t - \alpha_t \Delta f_{W'_t,J'_t}^{K,\mu}$, for all $t \leq T$, with respect to the sequences $S, S'$. Then the stability error $\delta_T$ satisfies the inequality of Lemma 2.*

We refer the reader to Appendix Section C.1, Theorem 14 for the formal statement of the result.[3] Through the Lipschitz condition of the loss and Lemma 8, we show that the mini-batch ZoSS enjoys the same generalization error bounds as in the case of single-query ZoSS ($m = 1$). As a consequence, the batch size does not affect the generalization error.

**Theorem 9 (Mini-batch ZoSS | Generalization Error)** *Let the loss function $f(\cdot, z)$ be L-Lipschitz and $\beta$-smooth (possibly nonconvex, possibly unbounded) for all $z \in \mathcal{Z}$. Then the bounds of Theorem 5 and Theorem 6 hold for the mini-batch ZoSS with iterate $W_{t+1} = W_t - \alpha_t \Delta f_{W_t,J_t}^{K,\mu}$, for all $t \leq T$ and any batch size $m \leq n$.*

---

[3]As in the single-query ($m = 1$) ZoSS, under the assumption of convex loss, the stability error of mini-batch ZoSS satisfies the inequality (46), Appendix B, Lemma 11.

By letting $K \to \infty$ and $c \to 0$, the generalization error bounds of mini-batch ZoSS reduce to those of mini-batch SGD, extending results of the single-query ($m = 1$) SGD that appeared in prior work [1]. Additionally, once $K \to \infty$, $c \to 0$ and $m = n$ we obtain generalization guarantees for full-batch GD. For the sake of clarity and completeness we provide dedicated stability and generalization analysis of full-batch GD in Appendix D, Corollary 15.

## 6   Discussion: Black-box Adversarial Attack Design and Future Work

A standard, well-cited example of ZoSS application is adversarial learning as considered in [5], when the gradient is not known for the adversary (for additional applications for instance federated/reinforcement learning, linear quadratic regulators; see also Section 1 for additional references). Notice that the algorithm in [5] is restrictive in the high dimensional regime since it requires $2d$ function evaluations per iteration. In contrast, ZoSS can be considered with any $K \geq 2$ functions evaluations (the trade-off is between accuracy and resource allocation, which is also controlled through $K$). If $K = d + 1$ evaluations are available we recover guarantees for the deterministic zeroth-order approaches (similar to [5]).

Retrieving a large number of function evaluations often is not possible in practice. When a limited amount of function evaluations is available, the adversary obtains the solution (optimal attack) with an optimization error that scales by a factor of $\sqrt{d/K}$, and the generalization error of the attack is of the order $\sqrt{T}/n$ under appropriate choices of the step-size, the smoothing parameter $\mu$ and $K$. Fine tuning of the these parameters might be useful in practice, but in general $K$ should be chosen as large as possible. In contrast, $\mu$ should be small and satisfy the inequality $\mu \leq cL\Gamma_K^d/n\beta(3+d)^{3/2}$ (Theorem 6). For instance, in practice $\mu$ is often chosen between $10^{-10}$ and $10^{-8}$ (or even lower) and the ZoSS algorithm remains (numerically) stable.

For neural networks with smooth activation functions [71–73], the ZoSS algorithm does not require the smoothness parameter $\beta$ to be necessarily known, however if $\beta$ is large then the guarantees of the estimated model would be pessimistic. To ensure that the learning procedure is successful, the adversary can approximate $\beta$ (since the loss is not known) by estimating the (largest eigenvalue of the) Hessian through the available function evaluations [74, Section 4.1].

Although the non-smooth (convex) loss setting lies out of the scope of this work, it is expected to inherit properties and rates of the SGD for non-smooth losses (at least for sufficiently small smoothing parameter $\mu$). In fact, [45, page 3, Table 1] developed upper and lower bounds for the SGD in the non-smooth case, and they showed that standard step-size choices provide vacuous stability bound. Due to these inherent issues of non-smooth (and often convex only cases), the generalization error analysis of ZoSS for non-smooth losses remains open. Finally, information-theoretic generalization error bounds of ZoSS can potentially provide further insight into the problem, due to the noisy updates of the algorithm, and consist part of future work.

## 7   Conclusion

In this paper, we characterized the generalization ability of black-box learning models. Specifically, we considered the Zeroth-order Stochastic Search (ZoSS) algorithm, which evaluates smoothed approximations of the unknown gradient of the loss by only relying on $K + 1$ loss evaluations. Under the assumptions of a Lipschitz and smooth (unknown) loss, we showed that the ZoSS algorithm achieves the same generalization error bounds as that of SGD, while the learning rate is slightly decreased compared to that of SGD. The efficient generalization ability of ZoSS, together with strong optimality results related to the optimization error by Duchi et al. [13], makes it a robust and powerful algorithm for a variety of black-box learning applications and problems.

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

## Acknowledgments and Disclosure of Funding

We would like to thank the four anonymous reviewers for providing valuable comments and suggestions, which have improved the presentation of the results and the overall quality of our paper.

Amin Karbasi acknowledges funding in direct support of this work from NSF (IIS-1845032), ONR (N00014- 19-1-2406), and the AI Institute for Learning-Enabled Optimization at Scale (TILOS).

