# A Proofs

We start by providing a lemma that is useful in the proofs of our main results.

**Lemma 10 (Variance Reduction)** *Let $\mathbf{U}_k \in \mathbb{R}^d, k \in \{1, 2\ldots, K\}$ be i.i.d standard Gaussian.[4] For every random vector $\mathbf{V} \in \mathbb{R}^d$ independent of all $\mathbf{U}_k, k \in \{1, 2\ldots, K\}$, it is true that*

$$\mathbb{E}\left[\left\|\frac{1}{K}\sum_{k=1}^{K}\langle\mathbf{V},\mathbf{U}_k\rangle\mathbf{U}_k - \mathbf{V}\right\|\Big|\mathbf{V}\right] \leq \sqrt{\frac{3d-1}{K}}\|\mathbf{V}\|. \tag{27}$$

Different versions of Lemma 10 appear already in prior works (see, e.g., [13, Proof of Corollary 2]. For completeness and clarity we provide a short proof of Lemma 10 below (Appendix, Section A.1).

## A.1 Proof of Lemma 10

For fixed $\mathbf{V} \in \mathbb{R}^d$, we have due to independence

$$\mathbb{E}\left[\left\|\frac{1}{K}\sum_{k=1}^{K}\langle\mathbf{V},\mathbf{U}_k\rangle\mathbf{U}_k - \mathbf{V}\right\|^2\right] = \frac{1}{K^2}\mathbb{E}\left[\left\|\sum_{k=1}^{K}\langle\mathbf{V},\mathbf{U}_k\rangle\mathbf{U}_k - \mathbf{V}\right\|^2\right]$$

$$= \frac{1}{K^2}\sum_{k=1}^{K}\mathbb{E}\left[\|\langle\mathbf{V},\mathbf{U}_k\rangle\mathbf{U}_k - \mathbf{V}\|^2\right]$$

$$= \frac{1}{K}\mathbb{E}\left[\|\langle\mathbf{V},\mathbf{U}_1\rangle\mathbf{U}_1 - \mathbf{V}\|^2\right].$$

Now, again due to independence

$$\mathbb{E}\left[\|\langle\mathbf{V},\mathbf{U}_1\rangle\mathbf{U}_1 - \mathbf{V}\|^2\right] = \mathbb{E}\left[\|\langle\mathbf{V},\mathbf{U}_1\rangle\mathbf{U}_1\|^2 - 2\langle\langle\mathbf{V},\mathbf{U}_1\rangle\mathbf{U}_1,\mathbf{V}\rangle + \|\mathbf{V}\|^2\right]$$

$$= \mathbb{E}\left[(\langle\mathbf{V},\mathbf{U}_1\rangle)^2\|\mathbf{U}_1\|^2\right] - 2\mathbb{E}\left[\langle\mathbf{V},\mathbf{U}_1\rangle\langle\mathbf{U}_1,\mathbf{V}\rangle\right] + \|\mathbf{V}\|^2$$

$$= \mathbf{V}^T\mathbb{E}\left[\mathbf{U}_1\mathbf{U}_1^T\|\mathbf{U}_1\|^2\right]\mathbf{V} - 2\mathbf{V}^T\mathbb{E}\left[\mathbf{U}_1\mathbf{U}_1^T\right]\mathbf{V} + \|\mathbf{V}\|^2$$

$$= \sum_{i=1}^{d}\mathbf{V}^T\mathbb{E}\left[\mathbf{U}_1\mathbf{U}_1^T\left(U_1^{(i)}\right)^2\right]\mathbf{V} - 2\|\mathbf{V}\|^2 + \|\mathbf{V}\|^2$$

$$\leq \sum_{i=1}^{d}3\mathbf{V}^T\mathbf{V} - \|\mathbf{V}\|^2$$

$$= (3d-1)\|\mathbf{V}\|^2.$$

Therefore,

$$\mathbb{E}\left[\left\|\frac{1}{K}\sum_{k=1}^{K}\langle\mathbf{V},\mathbf{U}_k\rangle\mathbf{U}_k - \mathbf{V}\right\|^2\right] \leq \frac{(3d-1)\|\mathbf{V}\|^2}{K}.$$

Thus, if $\mathbf{V}$ is random and independent of all $\mathbf{U}_k$'s, it follows that

$$\mathbb{E}\left[\left\|\frac{1}{K}\sum_{k=1}^{K}\langle\mathbf{V},\mathbf{U}_k\rangle\mathbf{U}_k - \mathbf{V}\right\|\Big|\mathbf{V}\right] \leq \sqrt{\mathbb{E}\left[\left\|\frac{1}{K}\sum_{k=1}^{K}\langle\mathbf{V},\mathbf{U}_k\rangle\mathbf{U}_k - \mathbf{V}\right\|^2\Big|\mathbf{V}\right]}$$

$$\leq \sqrt{\frac{3d-1}{K}\|\mathbf{V}\|^2}$$

$$= \sqrt{\frac{3d-1}{K}}\|\mathbf{V}\|,$$

and our claim is proved. □

---

[4]Similar bounds (i.e., $\mathcal{O}(\sqrt{d/K})$) hold for other distributions as well, e.g., when the $\mathbf{U}_k$'s are uniformly distributed (and independent) in $[-1, +1]$.

## A.2 Proof of Theorem 3

We start by observing that $1 - \eta - \alpha_t \beta \sqrt{\frac{3d-1}{K}} < 0$ since $\eta \geq 1$. The definition of $R_t$ and (18) give $R_t \leq (\eta + \alpha_t \beta \sqrt{(3d-1)/K}) \triangleq \tilde{R}_t$, and

$$\mathbb{E}[\delta_T | \mathcal{E}_{\delta_{t_0}}] \leq \left( \frac{2L}{n} \Gamma_K^d + \mu\beta(3+d)^{3/2} \right) \sum_{t=t_0+1}^{T} \alpha_t \prod_{j=t+1}^{T} \tilde{R}_j. \tag{28}$$

Recall that $\eta = 1 + \beta\alpha_t$ for general (nonconvex) losses (see [1]). Assuming that $\alpha_t \leq C/t\Gamma_K^d$ for all $t \leq T$, we have

$$\mathbb{E}[\delta_T | \mathcal{E}_{\delta_{t_0}}] \leq \left( \frac{2L}{n} \Gamma_K^d + \mu\beta(3+d)^{3/2} \right) \sum_{t=t_0+1}^{T} \alpha_t \prod_{j=t+1}^{T} \left( 1 + \alpha_j \beta \Gamma_K^d \right)$$

$$\leq \frac{C}{\Gamma_K^d} \left( \frac{2L}{n} \Gamma_K^d + \mu\beta(3+d)^{3/2} \right) \sum_{t=t_0+1}^{T} \frac{1}{t} \prod_{j=t+1}^{T} \left( 1 + \frac{C\beta}{j} \right)$$

$$\leq \frac{C}{\Gamma_K^d} \left( \frac{2L}{n} \Gamma_K^d + \mu\beta(3+d)^{3/2} \right) \sum_{t=t_0+1}^{T} \frac{1}{t} \prod_{j=t+1}^{T} \exp\left( \frac{C\beta}{j} \right) \tag{29}$$

$$\leq \frac{C(eT)^{\beta C}}{\Gamma_K^d} \left( \frac{2L}{n} \Gamma_K^d + \mu\beta(3+d)^{3/2} \right) \sum_{t=t_0+1}^{T} \frac{1}{t} \frac{1}{(t+1)^{\beta C}} \tag{30}$$

$$\leq \underbrace{\beta^{-1} \left( \Gamma_K^d \right)^{-1} \left( \frac{2L}{n} \Gamma_K^d + \mu\beta(3+d)^{3/2} \right)}_{D} \left( \left( \frac{eT}{t_0} \right)^{\beta C} - e^{\beta C} \right). \tag{31}$$

In the above, the inequality $1 + x \leq e^x$ gives (29), inequality (30) follows from the inequality $\sum_{j=t+1}^{T} 1/j \leq \log T - \log(t+1) + 1$, and inequality (31) comes from the next inequality and integral evaluation $\sum_{t=t_0+1}^{T} t^{-\beta C-1} \leq \int_{t=t_0}^{T} x^{-\beta C-1} dx = (\beta C)^{-1}(t_0^{-\beta C} - T^{-\beta C})$. We define $q \triangleq \beta C$ and find the value of $t_0$ that minimizes the right part of [75, Lemma 3.11][5]

$$\mathbb{E}\left[|f(W_T, z) - f(W_T', z)|\right] \leq \frac{t_0}{n} \sup_{w,z} f(w, z) + L\mathbb{E}[\delta_T | \mathcal{E}_{\delta_{t_0}}] \leq \frac{t_0}{n} + LD\left( \left( \frac{eT}{t_0} \right)^q - e^q \right), \tag{32}$$

which is $t_0^* = \min\{(qnLD)^{1/(q+1)} (eT)^{q/(q+1)}, T\}$. Then (32) gives

$$\mathbb{E}\left[|f(W_T, z) - f(W_T', z)|\right]$$

$$\leq \max\left\{ \frac{(qnLD)^{\frac{1}{q+1}} (eT)^{\frac{q}{q+1}}}{n}, \frac{1 + 1/q}{n} (qnLD)^{\frac{1}{q+1}} (eT)^{\frac{q}{q+1}} - LDe^q \right\}. \tag{33}$$

Choosing $\mu \leq cL\Gamma_K^d/n\beta(3+d)^{3/2}$ for some $c > 0$ in (33) proves our claim. $\qquad\square$

## A.3 Proof of Corollary 4

Denote by $W_0(\cdot)$ the Lambert function [76]. Through Theorem 3 and by replacing $C$ with $C\Gamma_K^d$ to recover the required learning rate, the generalization error is bounded as

$\epsilon_{\text{gen}}$

$$\leq \frac{1 + (\beta C \Gamma_K^d)^{-1}}{n}((2+c)CL^2)^{\frac{1}{1+\beta C\Gamma_K^d}} (\Gamma_K^d)^{\frac{1}{1+\beta C\Gamma_K^d}} (eT)^{\frac{\beta C\Gamma_K^d}{\beta C\Gamma_K^d+1}}$$

---

[5] [75, Lemma 3.11] applies to the ZoSS update rule (6) similar to SGD for nonnegative and $L$-Lipschitz losses. Note that $\mathbb{P}(\mathcal{I} \leq t_0) \leq t_0/n, t_0 \in \{0, \ldots, T\}$, and (32) comes from the Lipschitz assumption on the loss as $\mathbb{E}\left[|f(W_T, z) - f(W_T', z)|\right] \leq \mathbb{P}(\mathcal{I} \leq t_0)\mathbb{E}\left[|f(W_T, z) - f(W_T', z)||\mathcal{E}_{\delta_{t_0}}^c\right] + L\mathbb{E}[\delta_T | \mathcal{E}_{\delta_{t_0}}]$.

$$\leq \frac{1 + (\beta C \Gamma_K^d)^{-1}}{n}((2+c)CL^2)^{\frac{1}{1+\beta C \Gamma_K^d}} \left(\frac{1}{\beta C W_0\left(\frac{1}{\beta Ce}\right)}\right)^{\frac{1}{1+1/W_0\left(\frac{1}{\beta Ce}\right)}} (eT)^{\frac{\beta C \Gamma_K^d}{\beta C \Gamma_K^d + 1}} \tag{34}$$

$$\leq \frac{1 + (\beta C \Gamma_K^d)^{-1}}{n}((2+c)CL^2)^{\frac{1}{1+\beta C \Gamma_K^d}} \left(\frac{1}{W_0\left(1/e\right)}\right)^{\frac{1}{1+1/W_0\left(\frac{1}{e}\right)}} \max\{1, (\beta C)^{-1}\}(eT)^{\frac{\beta C \Gamma_K^d}{\beta C \Gamma_K^d + 1}}$$

$$\leq \frac{3}{2}\frac{1 + (\beta C \Gamma_K^d)^{-1}}{n}((2+c)CL^2)^{\frac{1}{1+\beta C \Gamma_K^d}} \max\{1, (\beta C)^{-1}\}(eT)^{\frac{\beta C \Gamma_K^d}{\beta C \Gamma_K^d + 1}} \tag{35}$$

$$\leq \frac{3}{2}\frac{1 + (\beta C)^{-1}}{n}((2+c)CL^2)^{\frac{1}{1+\beta C \Gamma_K^d}} \max\{1, (\beta C)^{-1}\}(eT)^{\frac{\beta C \Gamma_K^d}{\beta C \Gamma_K^d + 1}} \tag{36}$$

$$\leq \frac{3}{2}\frac{1 + (\beta C)^{-1}}{n} \max\{1, (2+c)CL^2\} \max\{1, (\beta C)^{-1}\}(eT)^{\frac{\beta C \Gamma_K^d}{\beta C \Gamma_K^d + 1}} \tag{37}$$

$$\leq \frac{3}{2}\frac{\left(1 + (\beta C)^{-1}\right)^2}{n} \left(1 + (2+c)CL^2\right)(eT)^{\frac{\beta C \Gamma_K^d}{\beta C \Gamma_K^d + 1}}$$

$$\leq \left(1 + (\beta C)^{-1}\right)^2 \left(1 + (2+c)CL^2\right)\frac{3Te}{2n}, \tag{38}$$

the maximization of $x^{1/(1+x\beta C)}$ gives (34), we find (35) by maximizing the term $(\beta C)^{-1/(1+1/W_0(1/xe))}$, and $W_0\left(1/xe\right)^{-1/(1+1/W_0(1/xe))}$, and by applying the inequality $W_0\left(1/e\right)^{-1/(1+1/W_0(1/xe))} \leq 3/2$. Inequality (36) holds since $\Gamma_K^d \geq 1$, we find (37) by maximizing the function $((2+c)CL^2)^{1/(1+\beta Cx)}$ for both cases $((2+c)CL^2) < 1$ and $((2+c)CL^2) \geq 1$. Finally, (38) holds for any value of $d \in \mathbb{N}$ and $K \in \mathbb{N}$ and gives the bound of the corollary. $\square$

## A.4 Proof of Theorem 5

We start by proving the first case of the Theorem for both convex and nonconvex loss.

**Proof of Theorem 5, First Case:** Let $\mathcal{C}$ denote the set of convex loss functions. Under the assumption $\alpha_t \leq C'/T$, and $\mu \leq cL\Gamma_K^d/(n\beta(3+d)^{3/2})$ Lemma 2 (nonconvex loss) and Lemma 11 (convex loss) give

$$\mathbb{E}[\delta_T | \mathcal{E}_{\delta_0}]$$

$$\leq \frac{(2+c)L\Gamma_K^d}{n} \sum_{t=1}^{T} \frac{C'}{T} \prod_{j=t+1}^{T}\left(1 + \frac{\beta C'}{T}\left(\mathbb{1}_{f(\cdot)\notin\mathcal{C}} + \sqrt{\frac{3d-1}{K}}\right)\right)$$

$$= \frac{(2+c)LC'\Gamma_K^d}{Tn} \sum_{t=1}^{T}\left(1 + \frac{\beta C'}{T}\left(\mathbb{1}_{f(\cdot)\notin\mathcal{C}} + \sqrt{\frac{3d-1}{K}}\right)\right)^{T-t}$$

$$= \frac{(2+c)LC'\Gamma_K^d}{Tn}\left(1 + \frac{\beta C'}{T}\left(\mathbb{1}_{f(\cdot)\notin\mathcal{C}} + \sqrt{\frac{3d-1}{K}}\right)\right)^{T} \sum_{t=1}^{T}\left(1 + \frac{\beta C'}{T}\left(\mathbb{1}_{f(\cdot)\notin\mathcal{C}} + \sqrt{\frac{3d-1}{K}}\right)\right)^{-t}$$

$$= \frac{(2+c)LC'\Gamma_K^d}{Tn}\frac{\left(1 + \frac{\beta C'}{T}\left(\mathbb{1}_{f(\cdot)\notin\mathcal{C}} + \sqrt{\frac{3d-1}{K}}\right)\right)^{T} - 1}{\left(1 + \beta\frac{C'}{T}\left(\mathbb{1}_{f(\cdot)\notin\mathcal{C}} + \sqrt{\frac{3d-1}{K}}\right)\right) - 1}$$

$$= \frac{(2+c)L\Gamma_K^d}{n}\frac{\left(1 + \frac{\beta C'}{T}\left(\mathbb{1}_{f(\cdot)\notin\mathcal{C}} + \sqrt{\frac{3d-1}{K}}\right)\right)^{T} - 1}{\beta\left(\mathbb{1}_{f(\cdot)\notin\mathcal{C}} + \sqrt{\frac{3d-1}{K}}\right)}$$

$$\leq \frac{(2+c)\,L\Gamma_K^d}{n} \frac{\exp\left(\beta C'\left(\mathbb{1}_{f(\cdot)\notin \mathcal{C}} + \sqrt{\frac{3d-1}{K}}\right)\right) - 1}{\beta\left(\mathbb{1}_{f(\cdot)\notin \mathcal{C}} + \sqrt{\frac{3d-1}{K}}\right)}.$$

If the loss is convex ($f(\cdot) \in \mathcal{C}$) and $\alpha_t \leq C'/T \leq 2/\beta$, the last display under the choice

$$C' = \frac{\log(1 + C\beta\sqrt{\frac{3d-1}{K}}/\Gamma_K^d)}{\beta\sqrt{\frac{3d-1}{K}}} \tag{39}$$

gives

$$\mathbb{E}[\delta_T|\mathcal{E}_{\delta_0}] \leq \frac{(2+c)\,L\Gamma_K^d}{n} \frac{\exp\left(\beta C'\sqrt{\frac{3d-1}{K}}\right) - 1}{\beta\sqrt{\frac{3d-1}{K}}} \leq \frac{(2+c)\,CL}{n}. \tag{40}$$

If the loss is nonconvex ($f(\cdot) \notin \mathcal{C}$) and $\alpha_t \leq C''/T$, then by choosing

$$C'' = \frac{\log(1 + C\beta)}{\beta\Gamma_K^d}, \tag{41}$$

we find

$$\mathbb{E}[\delta_T|\mathcal{E}_{\delta_0}] \leq \frac{(2+c)\,L\Gamma_K^d}{n} \frac{\exp\left(\beta C''\Gamma_K^d\right) - 1}{\beta\Gamma_K^d} \leq \frac{(2+c)\,CL}{n}. \tag{42}$$

The Lipschitz assumption $\mathbb{E}\left[|f(W_T, z) - f(W_T', z)|\right] \leq L\mathbb{E}[\delta_T] = L\mathbb{E}[\delta_T|\mathcal{E}_{\delta_0}]$ (as a consequence of $\mathbb{P}(\mathcal{I} \leq 0) = \mathbb{P}(\delta_0 > 0) = 0$) completes the proof for the first case of the theorem.

**Proof of Theorem 5, Second Case:** Lemma 2 (nonconvex loss) under the choice $t_0 = 0$ gives

$$\mathbb{E}[\delta_T|\mathcal{E}_{\delta_0}] \leq \left(\frac{2L}{n}\Gamma_K^d + \mu\beta(3+d)^{3/2}\right) \sum_{t=1}^{T} \alpha_t \prod_{j=t+1}^{T} \left(1 + \beta\alpha_j\left(1 + \sqrt{\frac{3d-1}{K}}\right)\right). \tag{43}$$

Under the assumption $\alpha_t \leq C/(T\Gamma_K^d)$, and $\mu \leq cL\Gamma_K^d/(n\beta(3+d)^{3/2})$ we find

$$\mathbb{E}[\delta_T|\mathcal{E}_{\delta_0}]$$

$$\leq (2+c)\,L\frac{\Gamma_K^d}{n} \sum_{t=1}^{T} \frac{C}{T\Gamma_K^d} \prod_{j=t+1}^{T} \left(1 + \frac{C\beta}{T}\right)$$

$$= (2+c)\,L\frac{\Gamma_K^d}{n} \frac{C}{T\Gamma_K^d} \sum_{t=1}^{T} \left(1 + \frac{C\beta}{T}\right)^{T-t}$$

$$= (2+c)\,L\frac{\Gamma_K^d}{n} \frac{C\left(1 + \frac{C\beta}{T}\right)^T}{T\Gamma_K^d} \sum_{t=1}^{T} \left(1 + \frac{C\beta}{T}\right)^{-t}$$

$$= \frac{(2+c)\,L}{n} \frac{C}{T} \frac{\left(1 + \frac{C\beta}{T}\right)^T - 1}{\left(1 + \frac{C\beta}{T}\right) - 1}$$

$$= \frac{(2+c)\,L}{n} \frac{\left(1 + \frac{C\beta}{T}\right)^T - 1}{\beta}. \tag{44}$$

The Lipschitz assumption $\mathbb{E}\left[|f(W_T, z) - f(W_T', z)|\right] \leq L\mathbb{E}[\delta_T] = L\mathbb{E}[\delta_T|\mathcal{E}_{\delta_0}]$ (as a consequence of $\mathbb{P}(\mathcal{I} \leq 0) = \mathbb{P}(\delta_0 > 0) = 0$) completes the proof. $\qquad\square$

## A.5 Proof of Theorem 6

Under the assumptions $\mu \leq cL\Gamma_K^d/(n\beta(3+d)^{3/2})$ and $\alpha_t \leq C/t\Gamma_K^d$, Lemma 2 gives

$$\mathbb{E}[\delta_T|\mathcal{E}_{\delta_0}]$$

$$\leq \left(\frac{2L}{n}\Gamma_K^d + \mu\beta(3+d)^{3/2}\right)\sum_{t=1}^{T}\alpha_t\prod_{j=t+1}^{T}\left(1+\beta\alpha_j\left(1+\sqrt{\frac{3d-1}{K}}\right)\right)$$

$$\leq \frac{\Gamma_K^d}{n}(2+c)L\sum_{t=1}^{T}\frac{C}{t\Gamma_K^d}\prod_{j=t+1}^{T}\left(1+\frac{C\beta}{j}\right)$$

$$\leq \frac{(2+c)L}{n}\sum_{t=1}^{T}\frac{C}{t}\exp\left(\sum_{j=t+1}^{T}\frac{C\beta}{j}\right)$$

$$\leq \frac{(2+c)L}{n}\sum_{t=1}^{T}\frac{C}{t}\exp\left(C\beta\log\left(\frac{eT}{t+1}\right)\right)$$

$$= \frac{C(2+c)L}{n}\sum_{t=1}^{T}\frac{1}{t}\left(\frac{eT}{t+1}\right)^{C\beta}$$

$$\leq \frac{C(eT)^{C\beta}(2+c)L}{n}\sum_{t=1}^{T}\frac{1}{t^{C\beta+1}}$$

$$\leq \frac{(eT)^{C\beta}(2+c)L}{n}\min\left\{\frac{C\beta+1}{\beta}, C\log(eT)\right\}. \tag{45}$$

The last inequality holds because of the inequalities $\sum_{t=1}^{T}t^{-C\beta-1} \leq \sum_{t=1}^{\infty}t^{-C\beta-1} \leq (C\beta+1)/C\beta$ and $\sum_{t=1}^{T}t^{-C\beta-1} \leq \sum_{t=1}^{T}1/t \leq \log(eT)$. Then the inequality $\mathbb{E}\left[|f(W_T,z) - f(W_T',z)|\right] \leq L\mathbb{E}[\delta_T] = L\mathbb{E}[\delta_T|\mathcal{E}_{\delta_0}]]$ (as a consequence of $\mathbb{P}(\mathcal{I} \leq 0) = \mathbb{P}(\delta_0 > 0) = 0$) completes the proof. $\square$

# B  Complementary Results

In this Section we provide complementary results and the corresponding proofs. The next result provides the equivalent bound of Lemma 2 for convex losses.

**Lemma 11 (ZoSS Stability Convex Loss)** *Assume that the loss function $f(\cdot, z)$ is $L$-Lipschitz, convex and $\beta$-smooth for all $z \in \mathcal{Z}$. Consider the ZO-SM algorithm* (6) *with parameters estimates $W_T$ and $W_T'$ for all the data-sets $S, S'$ respectively (that differ in exactly one entry). Then the discrepancy $\delta_T \triangleq \|W_T - W_T'\|$ under the event $\mathcal{E}_{\delta_{t_0}}$ satisfies the following inequality,*

$$\mathbb{E}[\delta_T|\mathcal{E}_{\delta_{t_0}}] \leq \left(\frac{2L}{n}\Gamma_K^d + \mu\beta(3+d)^{3/2}\right)\sum_{t=t_0+1}^{T}\alpha_t\prod_{j=t+1}^{T}\left(1+\beta\alpha_j\sqrt{\frac{3d-1}{K}}\right). \tag{46}$$

We prove Lemma 11 in parallel with Lemma 2 in Section 4.1 of the main part of the paper.

# C  ZoSS with Mini-Batch (Section 5)

For the stability analysis of mini-batch ZoSS, we similarly consider the sequences $S, S'$ that differ in one example. At each time $t$ we sample a batch $J_t \subset S$ ($J_t' \subset S'$) (and batch size $m \triangleq |J_t| = |J_t'|$) with replacement, or by considering random permutation of the samples and then sample the first $m$ examples. As a consequence in both cases $\mathbb{P}(J_t \neq J_t') = m/n$. Under the event $\{J_t \neq J_t'\}$, the sets $J_t, J_t'$ differ in one example $z_{i^*} \neq z_{i^*}'$ (for some $i^*$ without loss of generality), and $z_i = z_i'$ for any $z_i \in J_t$ and $i \in \{1, \ldots, m\} \setminus \{i^*\}$. Let $U_{k,i}^t \sim \mathcal{N}(0, I_d)$ be independent for all $k \in \{1, 2, \ldots, K\}$, $i \in \{1, 2, \ldots, m\}$ and $t \leq T$ and $\mu \in \mathbb{R}^+$. Recall the definition of the smoothed approximation and

update rule mapping of mini-batch ZoSS,

$$\Delta f_{w,J_t}^{K,\mu} \equiv \Delta f_{w,J_t}^{K,\mu,\mathbf{U}^t} \triangleq \frac{1}{mK} \sum_{i=1}^{m} \sum_{k=1}^{K} \frac{f(w + \mu U_{k,i}^t, z_{J_{t,i}}) - f(w, z_{J_{t,i}})}{\mu} U_{k,i}^t, \qquad (47)$$

$$\tilde{G}_{J_t}(w) \triangleq w - \alpha_t \Delta f_{w,J_t}^{K,\mu}, \quad \tilde{G}'_{J'_t}(w) \triangleq w - \alpha_t \Delta f_{w,J'_t}^{K,\mu}. \qquad (48)$$

For the stability error decomposition, we define the gradient based mappings $G_{J_t}(\cdot)$ and $G'_{J'_t}(\cdot)$ as

$$G_{J_t}(w) \triangleq w - \alpha_t \nabla f_{w,J_t}, \ \ G'_{J'_t}(w) \triangleq w - \alpha_t \nabla f_{w,J'_t}, \text{ and } \nabla f_{w,J_t} \triangleq \frac{1}{m} \sum_{z \in J_t} \nabla f(w, z) \qquad (49)$$

the iterate stability error of the mini-batch ZoSS $\tilde{G}_{J_t}(w) - \tilde{G}'_{J'_t}(w')$ at time $t$ and similarly to (5), we show that

$$\tilde{G}_{J_t}(w) - \tilde{G}'_{J'_t}(w') \propto \underbrace{G_{J_t}(w) - G'_{J'_t}(w')}_{\epsilon_{\text{GBstab}}^m} + \underbrace{\left[ \nabla f_{w,J_t} - \Delta f_{w,J_t}^{K,\mu} \right] + \left[ \nabla f_{w',J'_t} - \Delta f_{w',J'_t}^{K,\mu} \right]}_{\epsilon_{\text{est}}^m}. \qquad (50)$$

For the mini-batch case the derivation of the stability differs to that of (5) ($m = 1$).To analyze the error term $\epsilon_{\text{GBstab}}^m$ in the mini batch case, we derive (for the proof see Appendix, Section C.2) and apply the Mini-Batch SGD Growth Recursion for the mappings $G_{J_t}(\cdot), G'_{J'_t}(\cdot)$, that is an extension of [1, Lemma 2.4] and describes the growth recursion property of the SGD algorithm with mini batch.

**Lemma 12 (Mini-Batch SGD Growth Recursion)** *Fix arbitrary sequences of updates $\{G_{J_t}\}_{t=1}^T$ and $\{G'_{J'_t}\}_{t=1}^T$. Let $w_0 = w'_0$ be the starting point, $w_{t+1} = G_{J_t}(w_t)$ and $w'_{t+1} = G'_{J'_t}(w'_t)$ for any $t \in \{1, \ldots, T\}$. Then for any $t \geq 0$ the following recursion holds*

$$\|G_{J_t}(w_t) - G'_{J'_t}(w'_t)\| \leq \begin{cases} (1 + \beta\alpha_t)\|w_t - w'_t\| & \text{if } G_{J_t}(\cdot) = G'_{J'_t}(\cdot) \\ \left(1 + \frac{m-1}{m}\beta\alpha_t\right)\|w_t - w'_t\| + \frac{2}{m}L\alpha_t & \text{if } G_{J_t}(\cdot) \neq G'_{J'_t}(\cdot). \end{cases} \qquad (51)$$

The error $\epsilon_{\text{GBstab}}^m$ depends on the batch size $m$ at the event of different batch selection $\{J_t \neq J'_t\}$ as appears in Lemma 12. Additionally, the error $\epsilon_{\text{est}}^m$ breaks down into the errors $\epsilon_\mu, \epsilon_{d/K}^m$. Although $\epsilon_\mu$ is independent of $m$, $\epsilon_\mu \leq \mu\beta\mathbb{E}[\|U\|^3]$ ($U \in \mathbb{R}^d$ is standard normal), $\epsilon_{\text{est}}^m$ dependents on the batch size $m$ similarly to gradient based stability error $\epsilon_{\text{GBstab}}^m$. If the randomized algorithm (at time $t$) selects $J_t \neq J'_t$ then $\epsilon_{d/K}^m \leq 2\beta\alpha_t\sqrt{d/K}[(m-1)\|w - w'\| + 2L]/m$, else $\epsilon_{d/K}^m \leq 4\beta\alpha_t\|w - w'\|\sqrt{d/K}$. We provide a unified representation of the stability error $\tilde{G}_{J_t}(w) - \tilde{G}'_{J'_t}(w')$ in the Section (Mini-Batch ZoSS Growth Recursion, Lemma 13).

### C.1 Results for the ZoSS with Mini-Batch

We start by providing the growth recursion lemma for the ZoSS with mini batch.

**Lemma 13 (Mini-Batch ZoSS Growth Recursion)** *Consider the sequences of updates $\{\tilde{G}_{J_t}\}_{t=1}^T$ and $\{\tilde{G}'_{J_t}\}_{t=1}^T$ and $\mu \leq cL\Gamma_K^d/(n\beta(3+d)^{3/2})$. Let $w_0 = w'_0$ be the starting point, $w_{t+1} = \tilde{G}_{J_t}(w_t)$ and $w'_{t+1} = \tilde{G}'_{J_t}(w'_t)$ for any $t \in \{1, \ldots, T\}$. Then for any $w_t, w'_t \in \mathbb{R}^d$ and $t \geq 0$ the following recursion holds*

$$\mathbb{E}[\|\tilde{G}_{J_t}(w_t) - \tilde{G}'_{J_t}(w'_t)\|] \leq \begin{cases} \left(1 + \beta\alpha_t\Gamma_K^d\right)\delta_t + \frac{cL\alpha_t}{n}\Gamma_K^d & \text{if } \tilde{G}_{J_t}(\cdot) = \tilde{G}'_{J_t}(\cdot) \\ \left(1 + \frac{m-1}{m}\beta\alpha_t\Gamma_K^d\right)\delta_t + \frac{2L\alpha_t}{m}\Gamma_K^d + \frac{cL\alpha_t}{n}\Gamma_K^d & \text{if } \tilde{G}_{J_t}(\cdot) \neq \tilde{G}'_{J_t}(\cdot). \end{cases}$$

Our next result provides a stability guarantee on the difference of the mini-batch ZoSS outputs $W_T, W'_T$, that holds for any batch size $m$.

**Theorem 14 (Stability of ZoSS with Mini Batch | Nonconvex Loss)** *Assume that the loss function $f(\cdot, z)$ is L-Lipschitz and $\beta$-smooth for all $z \in \mathcal{Z}$. Consider the ZoSS with mini batch of size $m \in \{1, \ldots, n\}$, initial state $W_0 = W'_0$, iterates $W_t = \tilde{G}_{J_t}(W_t)$, $W'_t = \tilde{G}'_{J'_t}(W'_t)$ for $t > 0$, and with final-iterate estimates $W_T$ and $W'_T$ corresponding to the data-sets $S, S'$, respectively (that differ*

in exactly one entry). Then the discrepancy $\delta_T \triangleq \|W_T - W'_T\|$ under the event $\mathcal{E}_{\delta_{t_0}}$ and the choice $\mu \leq cL\Gamma_K^d/(n\beta(3+d)^{3/2})$ satisfies the inequality

$$\mathbb{E}[\delta_T | \mathcal{E}_{\delta_{t_0}}] \leq \frac{(2+c)L\Gamma_K^d}{n} \sum_{t=t_0+1}^{T} \alpha_t \prod_{j=t+1}^{T} \left(1 + \beta\alpha_j\Gamma_K^d\left(1 - \frac{1}{n}\right)\right). \tag{52}$$

We prove Theorem 14 in Appendix, Section C.3. Note that[6] $\mathbb{P}(\mathcal{I} \leq t_0) = 1 - (1 - m/n)^{t_0}$. By setting the free parameter $t_0 = 0$ and through the Lipschitz assumption we find the stability bound of the loss as $\mathbb{E}\left[|f(W_T, z) - f(W'_T, z)|\right] \leq L\mathbb{E}[\delta_T] = L\mathbb{E}[\delta_T | \mathcal{E}_{\delta_0}]]$. The last inequality and the solution of the recursion in (52) show that Theorem 5 and Theorem 6 hold for the ZoSS algorithm with mini batch, and any batch size $m \in \{1, \ldots, n\}$ as well.

## C.2 Proof of Lemma 12

Under the assumption of nonconvex losses we find the first part of the statement as

$$\|G_{J_t}(w_t) - G_{J_t}(w'_t)\| \leq \|w_t - w'_t\| + \frac{\alpha_t}{m}\left\| \sum_{z \in J_t} \nabla_w f(w,z)|_{w=w_t} - \sum_{z \in J_t} \nabla_w f(w,z)|_{w=w'_t} \right\|$$

$$\leq \|w_t - w'_t\| + \frac{\alpha_t}{m} \sum_{z \in J_t} \left\| \nabla_w f(w,z)|_{w=w_t} - \nabla_w f(w,z)|_{w=w'_t} \right\|$$

$$\leq \|w_t - w'_t\| + \frac{\alpha_t}{m} \sum_{z \in J_t} \beta\|w_t - w'_t\|$$

$$= (1 + \beta\alpha_t)\|w_t - w'_t\|. \tag{53}$$

Further define $J_t^{-i^*} \triangleq J_t \setminus \{z_{J_{t,i^*}}\}$ and $J_t'^{-i^*} \triangleq J_t' \setminus \{z'_{J'_{t,i^*}}\}$, and notice that $J_t^{-i^*} = J_t'^{-i^*}$ for any $t \leq T$ w.p. 1.

$$\|G_{J_t}(w_t) - G'_{J'_t}(w'_t)\|$$

$$= \left\| w_t - w'_t - \frac{\alpha_t}{m} \sum_{z \in J_t} \nabla_w f(w,z)|_{w=w_t} + \frac{\alpha_t}{m} \sum_{z' \in J'_t} \nabla_w f(w,z')|_{w=w'_t} \right\|$$

$$= \left\| \frac{1}{m} \sum_{z \in J_t^{-i^*}} (w_t - \alpha_t \nabla_w f(w,z)|_{w=w_t}) - \frac{1}{m} \sum_{z' \in J_t'^{-i^*}} \left(w'_t - \alpha_t \nabla_w f(w,z')|_{w=w'_t}\right) \right.$$

$$\left. + \frac{1}{m}\left(w_t - \alpha_t \nabla_w f(w, z_{J_{t,i^*}})|_{w=w_t}\right) - \frac{1}{m}\left(w'_t - \alpha_t \nabla_w f(w, z_{J'_{t,i^*}})|_{w=w'_t}\right) \right\|$$

$$= \left\| \frac{1}{m} \sum_{z \in J_t^{-i^*}} \underbrace{(w_t - \alpha_t \nabla_w f(w,z)|_{w=w_t})}_{G(w_t,z)} - \frac{1}{m} \sum_{z \in J_t^{-i^*}} \underbrace{(w'_t - \alpha_t \nabla_w f(w,z)|_{w=w'_t})}_{G(w'_t,z)} \right.$$

$$\left. + \frac{1}{m}\underbrace{(w_t - \alpha_t \nabla_w f(w, z_{J_{t,i^*}})|_{w=w_t})}_{G(w_t, z_{J_{t,i^*}})} - \frac{1}{m}\underbrace{(w'_t - \alpha_t \nabla_w f(w, z_{J'_{t,i^*}})|_{w=w'_t})}_{G'(w'_t, z_{J'_{t,i^*}})} \right\|$$

$$= \frac{1}{m}\left\| \sum_{z \in J_t^{-i^*}} (G(w_t,z) - G(w'_t,z)) + G(w_t, z_{J_{t,i^*}}) - G'(w'_t, z_{J'_{t,i^*}}) \right\|$$

$$\leq \frac{1}{m}\left\| \sum_{z \in J_t^{-i^*}} (G(w_t,z) - G(w'_t,z)) \right\| + \frac{1}{m}\|G(w_t, z_{J_{t,i^*}}) - G'(w'_t, z_{J'_{t,i^*}})\|$$

$$\leq \frac{1}{m} \sum_{z \in J_t^{-i^*}} \|G(w_t,z) - G(w'_t,z)\| + \frac{1}{m}\|G(w_t, z_{J_{t,i^*}}) - G'(w'_t, z_{J'_{t,i^*}})\|. \tag{54}$$

---

[6]Under the random selection rule $\mathbb{P}(\mathcal{I} \leq t_0) = 1 - \mathbb{P}(\cap_{i=1}^{t_0}\{\mathcal{I} \neq i\}) = 1 - \prod_{i=1}^{t_0} \mathbb{P}(\{\mathcal{I} \neq i\})$.

[1, Lemma 2.4] for nonconvex loss ($\eta = 1 + \beta\alpha_t$) gives

$$\|G(w_t, z) - G(w_t', z)\| \leq (1 + \beta\alpha_t)\delta_t, \tag{55}$$

$$\|G(w_t, z_{J_{t,i^*}}) - G'(w_t', z_{J_{t,i^*}'})\| \leq \delta_t + 2L\alpha_t. \tag{56}$$

By combining the last two together with (54) we find

$$
\begin{aligned}
\|G_{J_t}(w_t) - G'_{J_t'}(w_t')\| &\leq \frac{1}{m} \sum_{z \in J_t^{-i^*}} (1 + \beta\alpha_t)\delta_t + \frac{1}{m}(\delta_t + 2L\alpha_t) \\
&= \frac{m-1}{m}(1 + \beta\alpha_t)\delta_t + \frac{1}{m}(\delta_t + 2L\alpha_t) \\
&= \left(1 + \frac{m-1}{m}\beta\alpha_t\right)\delta_t + \frac{2}{m}L\alpha_t.
\end{aligned}
\tag{57}
$$

The last gives the second part of the recursion and completes the proof. □

### C.3 Proof of Lemma 13 and Theorem 14

First we provide the proof of Lemma 13, then we apply Lemma 13 to prove Theorem 14.

**Proof of Lemma 13** Consider the update rules under the event $\tilde{\mathcal{E}}_t \triangleq \{\tilde{G}_{J_t}(\cdot) \equiv \tilde{G}'_{J_t'}(\cdot)\}$ that occurs with probability $\mathbb{P}(\tilde{\mathcal{E}}_t) = 1 - m/n$ for all $t \leq T$. Similarly to (9) we find

$$
\begin{aligned}
&\tilde{G}_{J_t}(w_t) - \tilde{G}_{J_t}(w_t') \\
&= \underbrace{w_t - \frac{\alpha_t}{m}\sum_{i=1}^{m}\nabla_w f(w, z_{J_{t,i}})|_{w=w_t}}_{G_{J_t}(w_t)} - \underbrace{\left(w_t' - \frac{\alpha_t}{m}\sum_{i=1}^{m}\nabla_w f(w, z_{J_{t,i}})|_{w=w_t'}\right)}_{G'_{J_t}(w_t') \equiv G_{J_t}(w_t')} \\
&\quad - \frac{\alpha_t}{mK}\sum_{i=1}^{m}\sum_{k=1}^{K}\left(\frac{\mu}{2}U_k^{\mathrm{T}}\nabla_w^2 f(w, z_{J_{t,i}})|_{w=W_{k,t,i}^*}U_{k,i}^t\right)U_{k,i}^t \\
&\quad + \frac{\alpha_t}{mK}\sum_{i=1}^{m}\sum_{k=1}^{K}\left(\frac{\mu}{2}U_k^{\mathrm{T}}\nabla_w^2 f(w, z_{J_{t,i}})|_{w=W_{k,t,i}^{\dagger}}U_{k,i}^t\right)U_{k,i}^t \\
&\quad - \frac{\alpha_t}{m}\sum_{i=1}^{m}\left(\frac{1}{K}\sum_{k=1}^{K}\langle\nabla_w f(w, z_{J_{t,i}})|_{w=w_t} - \nabla_w f(w, z_{J_{t,i}})|_{w=w_t'}, U_{k,i}^t\rangle U_{k,i}^t \right. \\
&\qquad \left. - (\nabla_w f(w, z_{J_{t,i}})|_{w=w_t} - \nabla_w f(w, z_{J_{t,i}})|_{w=w_t'})\right).
\end{aligned}
\tag{58}
$$

Denote by $\mathbb{E}_{\mathbf{U}_t^{\otimes K \times m}}$ the expectation with respect to product measure of the random vectors $U_{k,i}^t \sim \mathcal{N}(0, I_d)$ for all $k \in \{1, 2, \ldots, K\}$, $i \in \{1, 2, \ldots, m\}$ and fixed $t \leq T$. Recall that $U_{k,i}^t$ are independent for all $k \in \{1, 2, \ldots, K\}$, $i \in \{1, 2, \ldots, m\}$ and $t \leq T$. Inequality (58) and triangle inequality give

$$
\begin{aligned}
&\mathbb{E}[\|\tilde{G}_{J_t}(w_t) - \tilde{G}_{J_t}(w_t')\|] \\
&\leq \|G_{J_t}(w_t) - G_{J_t}(w_t')\| \\
&\quad + \mathbb{E}_{\mathbf{U}_t^{\otimes K \times m}}\left[\left\|\frac{\alpha_t}{mK}\sum_{i=1}^{m}\sum_{k=1}^{K}\left(\frac{\mu}{2}U_k^{\mathrm{T}}\nabla_w^2 f(w, z_{J_{t,i}})|_{w=W_{k,t,i}^*}U_{k,i}^t\right)U_{k,i}^t\right\|\right] \\
&\quad + \mathbb{E}_{\mathbf{U}_t^{\otimes K \times m}}\left[\left\|\frac{\alpha_t}{mK}\sum_{i=1}^{m}\sum_{k=1}^{K}\left(\frac{\mu}{2}U_k^{\mathrm{T}}\nabla_w^2 f(w, z_{J_{t,i}})|_{w=W_{k,t,i}^{\dagger}}U_{k,i}^t\right)U_{k,i}^t\right\|\right] \\
&\quad + \mathbb{E}_{\mathbf{U}_t^{\otimes K \times m}}\left[\left\|\frac{\alpha_t}{m}\sum_{i=1}^{m}\left(\frac{1}{K}\sum_{k=1}^{K}\langle\nabla_w f(w, z_{J_{t,i}})|_{w=w_t} - \nabla_w f(w, z_{J_{t,i}})|_{w=w_t'}, U_{k,i}^t\rangle U_{k,i}^t\right.\right.
\end{aligned}
$$

$$- \left( \nabla_w f(w, z_{J_{t,i}})|_{w=w_t} - \nabla_w f(w, z_{J_{t,i}})|_{w=w'_t} \right) \right) \right\| \right]$$

$$\leq \|G_{J_t}(w_t) - G_{J_t}(w'_t)\|$$

$$+ \mathbb{E}_{\mathbf{U}_t^{\otimes K \times m}} \left[ \left\| \frac{\alpha_t}{mK} \sum_{i=1}^m \sum_{k=1}^K \left( \frac{\mu}{2} U_k^{\mathrm{T}} \nabla_w^2 f(w, z_{J_{t,i}})|_{w=W_{k,t,i}^*} U_{k,i}^t \right) U_{k,i}^t \right\| \right]$$

$$+ \mathbb{E}_{\mathbf{U}_t^{\otimes K \times m}} \left[ \left\| \frac{\alpha_t}{mK} \sum_{i=1}^m \sum_{k=1}^K \left( \frac{\mu}{2} U_k^{\mathrm{T}} \nabla_w^2 f(w, z_{J_{t,i}})|_{w=W_{k,t,i}^\dagger} U_{k,i}^t \right) U_{k,i}^t \right\| \right]$$

$$+ \frac{\alpha_t}{m} \sum_{i=1}^m \mathbb{E}_{\mathbf{U}_{t,i}^{\otimes K}} \left\| \left( \frac{1}{K} \sum_{k=1}^K \langle \nabla_w f(w, z_{J_{t,i}})|_{w=w_t} - \nabla_w f(w, z_{J_{t,i}})|_{w=w'_t}, U_{k,i}^t \rangle U_{k,i}^t \right. \right.$$

$$\left. \left. - \left( \nabla_w f(w, z_{J_{t,i}})|_{w=w_t} - \nabla_w f(w, z_{J_{t,i}})|_{w=w'_t} \right) \right) \right\|$$

$$\leq \|G_{J_t}(w_t) - G_{J_t}(w'_t)\|$$

$$+ \mathbb{E}_{\mathbf{U}_t^{\otimes K \times m}} \left[ \left\| \frac{\alpha_t}{mK} \sum_{i=1}^m \sum_{k=1}^K \left( \frac{\mu}{2} U_k^{\mathrm{T}} \nabla_w^2 f(w, z_{J_{t,i}})|_{w=W_{k,t,i}^*} U_{k,i}^t \right) U_{k,i}^t \right\| \right]$$

$$+ \mathbb{E}_{\mathbf{U}_t^{\otimes K \times m}} \left[ \left\| \frac{\alpha_t}{mK} \sum_{i=1}^m \sum_{k=1}^K \left( \frac{\mu}{2} U_k^{\mathrm{T}} \nabla_w^2 f(w, z_{J_{t,i}})|_{w=W_{k,t,i}^\dagger} U_{k,i}^t \right) U_{k,i}^t \right\| \right]$$

$$+ \frac{\alpha_t}{m} \sum_{i=1}^m \sqrt{\frac{3d-1}{K}} \|\nabla_w f(w, z_{J_{t,i}})|_{w=w_t} - \nabla_w f(w, z_{J_{t,i}})|_{w=w'_t}\| \tag{59}$$

$$\leq \|G_{J_t}(w_t) - G_{J_t}(w'_t)\| + \frac{2\alpha_t}{mK} \sum_{i=1}^m \sum_{k=1}^K \frac{\mu\beta}{2} \mathbb{E}_{\mathbf{U}_{t,i,k}} \left[ \|U_{k,i}^t\|^3 \right]$$

$$+ \frac{\alpha_t}{m} \sum_{i=1}^m \sqrt{\frac{3d-1}{K}} \|\nabla_w f(w, z_{J_{t,i}})|_{w=w_t} - \nabla_w f(w, z_{J_{t,i}})|_{w=w'_t}\| \tag{60}$$

$$\leq (1 + \beta\alpha_t)\delta_t + \frac{2\alpha_t}{mK} \sum_{i=1}^m \sum_{k=1}^K \frac{\mu\beta}{2} \mathbb{E}_{\mathbf{U}_{t,i,k}} \left[ \|U_{k,i}^t\|^3 \right] + \alpha_t \sqrt{\frac{3d-1}{K}} \beta\delta_t \tag{61}$$

$$\leq \left( 1 + \beta\alpha_t \Gamma_K^d \right) \delta_t + \mu\beta\alpha_t(3 + d)^{3/2}, \tag{62}$$

to find the inequality (59) we applied Lemma 10, inequality (60) comes from the triangle inequality and $\beta$-smoothness, to derive the inequality (61) we applied the $1 + \beta\alpha_t$-expansive property for the $G_{J_t}(\cdot)$ mapping (Lemma 12) and the $\beta$-smoothness of the loss function, finally the inequality (62) holds since the random vectors $U_{k,i}^t \sim \mathcal{N}(0, I_d)$ are i.i.d. and $\mathbb{E}[\|U_{k,i}^t\|] \leq (3 + d)^{3/2}$ for all $k \in \{1, 2, \ldots, K\}$, $i \in \{1, 2, \ldots, m\}$ and $t \leq T$. Under the choice $\mu \leq cL\Gamma_K^d/(n\beta(3 + d)^{3/2})$, (62) gives the first part the inequality in Lemma 13.

We continue by considering the event $\tilde{\mathcal{E}}_t^c \triangleq \{G_{J_t}(\cdot) \neq G'_{J'_t}(\cdot)\}$. Recall that $\tilde{\mathcal{E}}_t^c$ occurs with probability $\mathbb{P}(\tilde{\mathcal{E}}_t) = m/n$ for all $t \leq T$. Under the event $\tilde{\mathcal{E}}_t^c$ similarly to (13) we derive the difference

$$\tilde{G}_{J_t}(w_t) - \tilde{G}'_{J'_t}(w'_t)$$

$$= \underbrace{w_t - \frac{\alpha_t}{m} \sum_{i=1}^m \nabla_w f(w, z_{J_{t,i}})|_{w=w_t}}_{G_{J_t}(w_t)} - \underbrace{\left( w'_t - \frac{\alpha_t}{m} \sum_{i=1}^m \nabla_w f(w, z'_{J'_{t,i}})|_{w=w'_t} \right)}_{G'_{J_t}(w'_t)}$$

$$- \frac{\alpha_t}{mK} \sum_{i=1}^m \sum_{k=1}^K \left( \frac{\mu}{2} U_k^{\mathrm{T}} \nabla_w^2 f(w, z_{J_{t,i}})|_{w=W_{k,t,i}^*} U_{k,i}^t \right) U_{k,i}^t$$

$$
+ \frac{\alpha_t}{mK} \sum_{i=1}^{m} \sum_{k=1}^{K} \left( \frac{\mu}{2} U_k^{\mathrm{T}} \nabla_w^2 f(w, z'_{J'_{t,i}})|_{w=W_{k,t,i}^\dagger} U_{k,i}^t \right) U_{k,i}^t
$$

$$
- \frac{\alpha_t}{m} \sum_{i=1}^{m} \left( \frac{1}{K} \sum_{k=1}^{K} \langle \nabla_w f(w, z_{J_{t,i}})|_{w=w_t} - \nabla_w f(w, z'_{J'_{t,i}})|_{w=w'_t}, U_{k,i}^t \rangle U_{k,i}^t \right.
$$

$$
\left. - (\nabla_w f(w, z_{J_{t,i}})|_{w=w_t} - \nabla_w f(w, z'_{J'_{t,i}})|_{w=w'_t}) \right). \tag{63}
$$

By using the triangle inequality and Lemma 10 we get

$$
\mathbb{E}[\|\tilde{G}_{J_t}(w_t) - \tilde{G}'_{J'_t}(w'_t)\|]
$$

$$
\leq \|G_{J_t}(w_t) - G'_{J'_t}(w'_t)\|
$$

$$
+ \mathbb{E}_{\mathbf{U}_t^{\otimes K \times m}} \left[ \left\| \frac{\alpha_t}{mK} \sum_{i=1}^{m} \sum_{k=1}^{K} \left( \frac{\mu}{2} U_k^{\mathrm{T}} \nabla_w^2 f(w, z_{J_{t,i}})|_{w=W_{k,t,i}^*} U_{k,i}^t \right) U_{k,i}^t \right\| \right]
$$

$$
+ \mathbb{E}_{\mathbf{U}_t^{\otimes K \times m}} \left[ \left\| \frac{\alpha_t}{mK} \sum_{i=1}^{m} \sum_{k=1}^{K} \left( \frac{\mu}{2} U_k^{\mathrm{T}} \nabla_w^2 f(w, z'_{J'_{t,i}})|_{w=W_{k,t,i}^\dagger} U_{k,i}^t \right) U_{k,i}^t \right\| \right]
$$

$$
+ \mathbb{E}_{\mathbf{U}_t^{\otimes K \times m}} \left[ \left\| \frac{\alpha_t}{m} \sum_{i=1}^{m} \left( \frac{1}{K} \sum_{k=1}^{K} \langle \nabla_w f(w, z_{J_{t,i}})|_{w=w_t} - \nabla_w f(w, z'_{J'_{t,i}})|_{w=w'_t}, U_{k,i}^t \rangle U_{k,i}^t \right. \right. \right.
$$

$$
\left. \left. \left. - (\nabla_w f(w, z_{J_{t,i}})|_{w=w_t} - \nabla_w f(w, z'_{J'_{t,i}})|_{w=w'_t}) \right) \right\| \right]
$$

$$
\leq \|G_{J_t}(w_t) - G'_{J'_t}(w'_t)\| + \frac{2\alpha_t}{mK} \sum_{i=1}^{m} \sum_{k=1}^{K} \frac{\mu\beta}{2} \mathbb{E}_{\mathbf{U}_{t,i,k}} \left[ \|U_{k,i}^t\|^3 \right]
$$

$$
+ \frac{\alpha_t}{m} \sum_{i=1}^{m} \mathbb{E}_{\mathbf{U}_{t,i}^{\otimes K}} \left\| \left( \frac{1}{K} \sum_{k=1}^{K} \langle \nabla_w f(w, z_{J_{t,i}})|_{w=w_t} - \nabla_w f(w, z'_{J'_{t,i}})|_{w=w'_t}, U_{k,i}^t \rangle U_{k,i}^t \right. \right.
$$

$$
\left. \left. - (\nabla_w f(w, z_{J_{t,i}})|_{w=w_t} - \nabla_w f(w, z'_{J'_{t,i}})|_{w=w'_t}) \right) \right\|
$$

$$
\leq \|G_{J_t}(w_t) - G'_{J'_t}(w'_t)\| + \frac{2\alpha_t}{mK} \sum_{i=1}^{m} \sum_{k=1}^{K} \frac{\mu\beta}{2} \mathbb{E}_{\mathbf{U}_{t,i,k}} \left[ \|U_{k,i}^t\|^3 \right]
$$

$$
+ \frac{\alpha_t}{m} \sum_{i=1}^{m} \sqrt{\frac{3d-1}{K}} \|\nabla_w f(w, z_{J_{t,i}})|_{w=w_t} - \nabla_w f(w, z'_{J'_{t,i}})|_{w=w'_t}\| \tag{64}
$$

$$
= \|G_{J_t}(w_t) - G'_{J'_t}(w'_t)\| + \frac{2\alpha_t}{mK} \sum_{i=1}^{m} \sum_{k=1}^{K} \frac{\mu\beta}{2} \mathbb{E}_{\mathbf{U}_{t,i,k}} \left[ \|U_{k,i}^t\|^3 \right]
$$

$$
+ \sqrt{\frac{3d-1}{K}} \frac{\alpha_t}{m} \sum_{i=1, i \neq i^*}^{m} \|\nabla_w f(w, z_{J_{t,i}})|_{w=w_t} - \nabla_w f(w, z'_{J'_{t,i}})|_{w=w'_t}\|
$$

$$
+ \sqrt{\frac{3d-1}{K}} \frac{\alpha_t}{m} \|\nabla_w f(w, z_{J_{t,i^*}})|_{w=w_t} - \nabla_w f(w, z'_{J'_{t,i^*}})|_{w=w'_t}\|
$$

$$
= \|G_{J_t}(w_t) - G'_{J'_t}(w'_t)\| + \frac{2\alpha_t}{mK} \sum_{i=1}^{m} \sum_{k=1}^{K} \frac{\mu\beta}{2} \mathbb{E}_{\mathbf{U}_{t,i,k}} \left[ \|U_{k,i}^t\|^3 \right]
$$

$$
+ \sqrt{\frac{3d-1}{K}} \frac{\alpha_t}{m} \sum_{i=1, i \neq i^*}^{m} \|\nabla_w f(w, z_{J_{t,i}})|_{w=w_t} - \nabla_w f(w, z_{J_{t,i}})|_{w=w'_t}\|
$$

$$+ \sqrt{\frac{3d-1}{K}} \frac{\alpha_t}{m} \|\nabla_w f(w, z_{J_{t,i^*}})|_{w=w_t} - \nabla_w f(w, z'_{J'_{t,i^*}})|_{w=w'_t}\| \tag{65}$$

$$\leq \|G_{J_t}(w_t) - G'_{J'_t}(w'_t)\| + \frac{2\alpha_t}{mK} \sum_{i=1}^{m} \sum_{k=1}^{K} \frac{\mu\beta}{2} \mathbb{E}_{\mathbf{U}_{t,i,k}} \left[\|U_{k,i}^t\|^3\right]$$

$$+ \sqrt{\frac{3d-1}{K}} \frac{\alpha_t}{m}(m-1)\beta\delta_t + \sqrt{\frac{3d-1}{K}} \frac{\alpha_t}{m} 2L \tag{66}$$

$$\leq \left(1 + \frac{m-1}{m}\beta\alpha_t\right)\delta_t + \frac{2}{m}L\alpha_t + \mu\beta\alpha_t(3+d)^{3/2}$$

$$+ \sqrt{\frac{3d-1}{K}} \frac{\alpha_t}{m}(m-1)\beta\delta_t + \sqrt{\frac{3d-1}{K}} \frac{\alpha_t}{m} 2L \tag{67}$$

$$= \left(1 + \frac{m-1}{m}\beta\alpha_t\Gamma_K^d\right)\delta_t + \frac{2L\alpha_t}{m}\Gamma_K^d + \mu\beta\alpha_t(3+d)^{3/2}, \tag{68}$$

we find the inequality (64) by applying Lemma 10, the inequality (65) holds since $z_{J_{t,i}} = z'_{J'_{t,i}}$ for any $i \neq i^*$, we find (66) by using the triangle inequality and $\beta$-smoothness (for $i \neq i^*$) and $L-$Lipschitz condition to bound the norm of the gradients $\nabla_w f(w, z_{J_{t,i^*}})|_{w=w_t}$ and $\nabla_w f(w, z'_{J'_{t,i^*}})|_{w=w'_t}$. In (67) we apply Lemma 12 to bound the quantity $\|G_{J_t}(w_t) - G'_{J_t}(w'_t)\|$. Under the selection of $\mu \leq cL\Gamma_K^d/(n\beta(3+d)^{3/2})$, Eq. (68) gives the second part of the inequality in Lemma 13. $\qquad\square$

**Proof of Theorem 14**   We apply Lemma 13 to get

$$\mathbb{E}[\delta_{t+1}|\mathcal{E}_{\delta_{t_0}}]$$

$$= \mathbb{P}(\mathcal{E}_t)\mathbb{E}[\delta_{t+1}|\mathcal{E}_t, \mathcal{E}_{\delta_{t_0}}] + \mathbb{P}(\mathcal{E}_t^c)\mathbb{E}[\delta_{t+1}|\mathcal{E}_t^c, \mathcal{E}_{\delta_{t_0}}]$$

$$= \left(1 - \frac{m}{n}\right)\mathbb{E}[\delta_{t+1}|\mathcal{E}_t, \mathcal{E}_{\delta_{t_0}}] + \frac{m}{n}\mathbb{E}[\delta_{t+1}|\mathcal{E}_t^c, \mathcal{E}_{\delta_{t_0}}]$$

$$= \left(1 - \frac{m}{n}\right)\left(\left(1 + \beta\alpha_t\Gamma_K^d\right)\mathbb{E}[\delta_t|\mathcal{E}_{\delta_{t_0}}] + \frac{cL\alpha_t}{n}\Gamma_K^d\right)$$

$$+ \frac{m}{n}\left(1 + \frac{m-1}{m}\beta\alpha_t\Gamma_K^d\right)\mathbb{E}[\delta_t|\mathcal{E}_{\delta_{t_0}}] + \frac{m}{n}\frac{2L\alpha_t}{m}\Gamma_K^d + \frac{m}{n}\frac{cL\alpha_t}{n}\Gamma_K^d$$

$$= \left(1 + \beta\alpha_t\Gamma_K^d\left(1 - \frac{1}{n}\right)\right)\mathbb{E}[\delta_t|\mathcal{E}_{\delta_{t_0}}] + \frac{2L\alpha_t}{n}\Gamma_K^d + \frac{cL\alpha_t}{n}\Gamma_K^d. \tag{69}$$

The last display characterizes the general case of nonconvex loss and coincides with the inequality (17) (since $\eta = 1 + \beta\alpha_t$). As a consequence the solution of the recursion (69) is

$$\mathbb{E}[\delta_T|\mathcal{E}_{\delta_{t_0}}] \leq \frac{(2+c)L\Gamma_K^d}{n} \sum_{t=t_0+1}^{T} \alpha_t \prod_{j=t+1}^{T} \left(1 + \beta\alpha_j\Gamma_K^d\left(1 - \frac{1}{n}\right)\right). \tag{70}$$

The last display completes the proof. $\qquad\square$

## D   Full-Batch GD

As a byproduct of our analysis we derive generalization error bounds for the full-batch gradient decent. Although our results reduce to the full-batch GD by a direct calculation of the limits $c \to 0$, $K \to \infty$ and setting the batch size $m$ equal to $n$, we separately prove generalization error bounds for the full-batch GD for clarity.

**Corollary 15** *[**Stability and Generalization Error of Full-Batch GD | Nonconvex Loss**] Assume that the loss function $f(\cdot, z)$ is L-Lipschitz and $\beta$-smooth for all $z \in \mathcal{Z}$. Consider the (deterministic) full-batch GD algorithm, initial state $W_0 = W'_0$, iterates $W_t = G_S(W_t)$, $W'_t = G'_{S'}(W'_t)$ for $t > 0$, and with final-iterate estimates $W_T$ and $W'_T$ corresponding to the data-sets $S, S'$, respectively (that differ in exactly one entry). Then the discrepancy $\delta_T \triangleq \|W_T - W'_T\|$ satisfies the inequality*

$$\delta_T \leq \frac{2L}{n} \sum_{t=1}^{T} \alpha_t \prod_{j=t+1}^{T} \left(1 + \frac{n-1}{n}\alpha_j\beta\right). \tag{71}$$

*Further if $\alpha_t \leq C/t$ for any $t > 0$ and some $C > 0$ then*

$$|\epsilon_{gen}| = |\mathbb{E}_S[\mathbb{E}_z[f(W_T, z)] - \frac{1}{n}\sum_{z\in S} f(W_T, z)]| \leq \frac{2L^2 (eT)^{C\beta}}{n} \min\{C + \beta^{-1}, C\log(eT)\}. \quad (72)$$

The proof of Corollary 15 follows.

**Proof of Corollary 15 (Full-Batch GD)** In the case of full-batch GD the algorithm is deterministic and we assume that $z_1, z_2, \ldots, z_i, \ldots, z_n, z'_i$ are i.i.d. and define $S \triangleq (z_1, z_2, \ldots, z_i, \ldots, z_n)$ and $S' \triangleq (z_1, z_2, \ldots, z'_i, \ldots, z_n)$, $W_0 = W'_0$, the updates for any $t \geq 1$ are

$$W_{t+1} = W_t - \frac{\alpha_t}{n} \sum_{j=1}^{n} \nabla f(W_t, z_j), \quad (73)$$

$$W'_{t+1} = W'_t - \frac{\alpha_t}{n} \sum_{j=1, j\neq i}^{n} \nabla f(W'_t, z_j) - \frac{\alpha_t}{n}\nabla f(W'_t, z'_i). \quad (74)$$

Then for any $t \geq 1$

$$\delta_{t+1}$$
$$\leq \delta_t + \frac{\alpha_t}{n}\left\| \sum_{j=1, j\neq i}^{n} \nabla f(W_t, z_j) - \nabla f(W'_t, z_j)\right\| + \frac{\alpha_t}{n}\|\nabla f(W_t, z_i) - \nabla f(W'_t, z'_i)\|$$
$$\leq \delta_t + \frac{\alpha_t(n-1)}{n}\beta\delta_t + \frac{2L\alpha_t}{n}$$
$$= \left(1 + \frac{(n-1)}{n}\beta\alpha_t\right)\delta_t + \frac{2L\alpha_t}{n}.$$

Then by solving the recursion we find

$$\delta_T \leq \frac{2L}{n}\sum_{t=1}^{T}\alpha_t \prod_{j=t+1}^{T}\left(1 + \frac{n-1}{n}\alpha_j\beta\right). \quad (75)$$

Under the choice $\alpha_t \leq C/t$ the last display gives

$$\delta_T \leq \frac{2L}{n}\sum_{t=1}^{T}\frac{C}{t}\prod_{j=t+1}^{T}\left(1 + \frac{n-1}{n}\frac{C}{j}\beta\right)$$
$$\leq \frac{2L}{n}\sum_{t=1}^{T}\frac{C}{t}\prod_{j=t+1}^{T}\left(1 + \frac{C}{j}\beta\right)$$
$$\leq \frac{2L}{n}\sum_{t=1}^{T}\frac{C}{t}\prod_{j=t+1}^{T}\exp\left(\frac{C}{j}\beta\right)$$
$$= \frac{2L}{n}\sum_{t=1}^{T}\frac{C}{t}\exp\left(\sum_{j=t+1}^{T}\frac{C}{j}\beta\right)$$
$$\leq \frac{2L}{n}\sum_{t=1}^{T}\frac{C}{t}\exp\left(C\beta\left(1 + \log\frac{T}{t+1}\right)\right)$$
$$= \frac{2L(eT)^{C\beta}}{n}\sum_{t=1}^{T}\frac{C}{t}\frac{1}{(t+1)^{C\beta}}$$
$$\leq \frac{2L(eT)^{C\beta}}{n}\sum_{t=1}^{T}\frac{C}{t^{C\beta+1}}$$

$$\leq \frac{2CL\,(eT)^{C\beta}}{n}\min\left\{\frac{C\beta+1}{C\beta}, \log(eT)\right\}. \tag{76}$$

Then

$$|\mathbb{E}_S[R_S(A_S) - R(A_S)]| = |\mathbb{E}_{S,z_i'}[f(W_T, z_i') - f(W_T', z_i')]| \tag{77}$$
$$\leq \mathbb{E}_{S,z_i'}[|f(W_T, z_i') - f(W_T', z_i')|]$$
$$\leq L\mathbb{E}_{S,z_i'}\|W_T - W_T'\| \tag{78}$$
$$\leq \frac{2L^2\,(eT)^{C\beta}}{n}\min\left\{C + \beta^{-1}, C\log(eT)\right\}. \tag{79}$$

In the above, Eq. (77) follows from [17, Lemma 7], the inequality (78) holds under the Lipschitz property of the loss $f(\cdot, z)$ for any $z$. Finally, we find the last inequality (79) by applying the bound in (76). □

## E  Excess Risk

Define the time average parameters as output of the algorithm

$$\bar{W}_T = \frac{1}{\sum_{t=1}^T \alpha_t}\sum_{t=1}^T \alpha_t W_t, \tag{80}$$

then

$$\mathbb{E}[\|\bar{W}_T - \bar{W}_T'\| | \mathcal{E}_{\delta_{t_0}}] \leq \frac{1}{\sum_{t=1}^T \alpha_t}\sum_{t=1}^T \alpha_t \mathbb{E}[\|W_t - W_t'\| | \mathcal{E}_{\delta_{t_0}}] \tag{81}$$

$$= \frac{1}{\sum_{t=1}^T \alpha_t}\sum_{t=1}^T \alpha_t \mathbb{E}[\delta_t | \mathcal{E}_{\delta_{t_0}}] \tag{82}$$

$$\leq \frac{1}{\sum_{t=1}^T \alpha_t}\sum_{t=1}^T \alpha_t \mathbb{E}[\delta_T | \mathcal{E}_{\delta_{t_0}}] = \mathbb{E}[\delta_T | \mathcal{E}_{\delta_{t_0}}]. \tag{83}$$

The L-Lipschitz property of the loss and the inequality (83) give

$$|\bar{\epsilon}_{\text{gen}}| \triangleq |\mathbb{E}[f(\bar{W}_T, z_i') - f(\bar{W}_T', z_i')]| \leq \mathbb{E}[|f(\bar{W}_T, z_i') - f(\bar{W}_T', z_i')|] \leq L\mathbb{E}[\delta_T | \mathcal{E}_{\delta_{t_0}}]. \tag{84}$$

Additionally, for any convex loss it is true that,

$$\bar{\epsilon}_{\text{opt}} \triangleq \mathbb{E}[R(\bar{W}_T)] - R(w^*) \leq \frac{1}{\sum_{t=1}^T \alpha_t}\sum_{t=1}^T \alpha_t\left(\mathbb{E}[R(W_t)] - R(w^*)\right)$$

$$\leq \frac{1}{\sum_{t=1}^T \alpha_t}\left(\frac{1}{2}\|W_0 - W^*\|^2 + \frac{d+4}{2}L\sum_{t=1}^T \alpha_t^2\right). \tag{85}$$

If $\|W_0 - W^*\|^2 \leq R$, $K = 1$, then we may choose

$$\alpha_t = \frac{CR}{L\sqrt{3d-1}t}. \tag{86}$$

From (85) and (86) we find

$$\bar{\epsilon}_{\text{opt}} \leq \frac{L\sqrt{3d-1}}{CR\log(T+1)}\left(\frac{R^2}{2} + \frac{d+4}{2}L\left(\frac{C^2R^2}{L^2(3d-1)}\right)\frac{\pi^2}{6}\right)$$

$$\leq \frac{RL\sqrt{3d-1}}{2C\log(T+1)}\left(1 + \frac{C^2}{L}\right). \tag{87}$$

Further, inequality (84), the choice of learning rate in (86) and Lemma 11 give

$$|\bar{\epsilon}_{\text{gen}}| \leq \frac{1 + \sqrt{3d-1}}{\sqrt{3d-1}}\frac{(eT)^{CR\beta/L}(2+c)L^2}{n}\min\left\{\frac{CR\beta/L+1}{\beta}, \frac{CR}{L}\log(eT)\right\} \tag{88}$$

$$\leq \frac{2(eT)^{CR\beta/L}(2+c)L^2}{n} \min\left\{\frac{CR\beta/L+1}{\beta}, \frac{CR}{L}\log(eT)\right\}. \tag{89}$$

If $C \leq L/2R\beta$, then

$$|\bar{\epsilon}_{\text{gen}}| \leq \frac{2\sqrt{eT}(2+c)L^2}{n} \min\left\{\frac{3}{2\beta}, \frac{1}{2\beta}\log(eT)\right\} \leq \frac{3\sqrt{eT}(2+c)L^2/\beta}{n}. \tag{90}$$

We conclude that

$$\bar{\epsilon}_{\text{excess}} \leq |\bar{\epsilon}_{\text{gen}}| + \bar{\epsilon}_{\text{opt}} \leq \frac{3\sqrt{eT}(2+c)L^2/\beta}{n} + \frac{\sqrt{3d-1}R^2\beta}{\log(T+1)}\left(1 + \frac{L}{4R^2\beta^2}\right). \tag{91}$$

Similarly, by using Lemma 10 and the optimization error derivation from prior works [13], we find the corresponding bound for $K \geq 1$ function evaluations,

$$\bar{\epsilon}_{\text{excess}} \leq \frac{3\sqrt{eT}(2+c)L^2/\beta}{n} + \frac{\left(1 + \sqrt{\frac{3d-1}{K}}\right)R^2\beta}{\log(T)}\left(1 + \frac{L}{4R^2\beta^2}\right). \tag{92}$$