# OpenReview forum: "Black-Box Generalization: Stability of Zeroth-Order Learning"
_NeurIPS.cc/2022/Conference — NeurIPS 2022 Accept_

### Official Review · Reviewer_HbvR · 2022-06-15

**Rating:** 6
**Confidence:** 3
**Soundness:** 3 good
**Presentation:** 3 good
**Contribution:** 3 good

**Summary:**

This paper provides generalization bounds for the ZoSS algorithm, based on the analysis of algorithmic stability. The bounds match the previous work of HRS16 on SGD. The main technical ingredient is a recursion lemma which controls how fast the expected distance between two ZoSS instances grow, when they differ on only one example. This result is then extended to a few different scenarios.

**Questions:**

What's 'high-probability analysis' (line 43)? Is it a typo?

Is it possible to achieve similar results for one-point zeroth-order algorithms? For example, the 'gradient descent without a gradient' method of FKM04.

**Limitations:**

The authors addressed some limitations

**Strengths And Weaknesses:**

The method is similar to the analysis of SGD (HRS16) in spirit, and the achieved bound is not very surprising. Still, it's good to see similar generalization bounds can be obtained for zeroth-order optimization (though it shares the same drawbacks of HRS16 as well, such as the linear dependence on $T$).

The recursion lemma is interesting.

The writing is good and the paper is easy-to-follow in general.

The title seems a bit unclear and misleading to me. I suggest change it to be more specific to avoid confusion (like 'generalization bounds for zeroth-order optimization via algorithmic stability'?). The analysis also applies only to ZoSS algorithms and doesn't seem to be able to cover all zeroth-order algorithms.

To conclude, the results of this paper are solid and meaningful in that it provides the first generalization analysis for zeroth-order optimization. But it's felt that there isn't enough technical depth (incremental over HRS16) and the paper is a bit over-selling.

---

> ### Author Response · Authors · 2022-08-01
> **Response to Reviewer HbvR**
>
>
> We would like to thank the Reviewer for their effort reading our paper, and for their comments, which we address as follows.
>
> We consider to change the title as the reviewer suggested in an upcoming version of the paper. While our analysis is dedicated to ZoSS as the reviewer noticed, we recall that given $K\geq 2$ function evaluations ZoSS has been proved to be optimal in [13]. This makes ZoSS the main/premier algorithm for black-box learning.
>
> Q.1 ''There isn't enough technical depth (incremental over HRS16) and the paper is a bit over-selling.''
>
>  A.1 We are not aware of any prior work or analysis regarding the generalization bound for zeroth order optimization method, even though from the optimization perspective there is a large body of work. Moreover, our result build on top of HRS16 in the following non-trivial way: we start with the decomposition of the ZoSS update into an gradient part and an approximation error such that no additional assumptions are needed (similar to those of SGD) in the analysis. Then we proceed with the derivation of the ZoSS growth recursion with meaningful stability guarantees. Specifically, we show that ZoSS satisfies uniform algorithmic stability even for small $\mu$, and we characterize the dependence of the growth with respect to the parameters of interest, including the number of function evaluations, dimensionality, number of samples and iterations. Finally, our results include ZoSS generalization error bounds that also consistently extend guarantees in prior works including unbounded losses, varying batch sizes, and recover the SGD bounds in the full-information regime ($K\rightarrow \infty$). We would appreciate if the reviewer can point which part of the paper is incremental and overselling so that we can properly address them.
>
> Q.2 What's 'high-probability analysis' (line 43)?
>
>  A.2 This typo will be addressed accordingly.
>
> Q.3 Is it possible to achieve similar results for one-point zeroth-order algorithms? For example, the 'gradient descent without a gradient' method of FKM04.
>
> A.3 For one-point zeroth-order methods the algorithm becomes unstable and the growth recursion will involve an extra term of the order $\mathcal{O}(1/\mu)$. In contrast, ZoSS provides the required stability for generalization.
>
> A brief derivation of the stability analysis in the one point zeroth-order update follows under the event $\mathcal{E}_t$, while the event $\mathcal{E}^c_t$ can be analyzed (for bounded losses) in a similar way.
>
> Under the event $\mathcal{E}_t$, the derivation of the growth recursion for one-point zeroth-order (for comparison see also equality (9) which is the ZoSS equivalent in the paper) gives
>
>
> $$ \begin{align}
>   &w_\{t+1 \} - w'_\{ t+1\} \\\\
>   & =  w_t - w'_t - \frac{\alpha_t}{K}\sum^K_\{k=1\} \frac{1}{\mu}  (f(w_t ,z_\{i_t\})- f(w'_t ,z_\{i_t\}))U^t_k -   \frac{\alpha_t}{K}\sum^K_\{k=1\} \left( (\nabla f(w_t ,z_\{i_t\})-\nabla f(w'_t ,z_\{i_t\})\right)^\text{T} \cdot U^t_k  )  U^t_k \\\\
> & \qquad - \frac{\alpha_t}{K}\sum^K_\{k=1\} \left( \frac{\mu}{2} (U^{t}_k)^\text{T} \nabla^2_\{w\}f(w ,z_\{i_t\})|_\{w=W^*_\{k,t\}\} U^t_k \right) U^t_k    + \frac{\alpha_t}{K}\sum^K_\{k=1\} \left( \frac{\mu}{2} (U^{t}_k )^\text{T} \nabla^2_\{w\}f(w ,z_\{i_t\})|_\{w=W^\dagger_\{k,t\}\} U^t_k\right) U^t_k \\\\
> & = w_t-\alpha_t  \nabla f(w_t ,z_\{i_t\}) -  \left( w'_t  -\alpha_t\nabla f(w'_t ,z_\{i_t\})\right) - \frac{\alpha_t}{K}\sum^K_\{k=1\} \frac{1}{\mu} (f(w_t ,z_\{i_t\})- f(w'_t ,z_\{i_t\}))U^t_k \\\\
> & \qquad - \frac{\alpha_t}{K}\sum^K_\{k=1\}  \left( \frac{\mu}{2} ( U^{t}_k )^\text{T} \nabla^2_\{w\}f(w ,z_\{i_t\})|_\{w=W^*_\{k,t\}\} U^t_k \right) U^t_k  + \frac{\alpha_t}{K}\sum^K_\{k=1\} \left( \frac{\mu}{2} ( U^{t}_k )^\text{T} \nabla^2_\{w\}f(w ,z_\{i_t\})|_\{w=W^\dagger_\{k,t\}\} U^t_k\right) U^t_k \\\\
> & \qquad  -   \alpha_t\bigg( \frac{1}{K}\sum^K_\{k=1\}\left( (\nabla f(w_t ,z_\{i_t\})-\nabla f(w'_t ,z_\{i_t\}))^\text{T} \cdot U^t_k \right) U^t_k- ( \nabla f(w_t ,z_\{i_t\})-\nabla f(w'_t ,z_\{i_t\}))\bigg)
> \end{align}$$
>
>
> After applying triangle inequality the one-point extra term contributes as
>
> $$    \mathbb{E}_{\mathbf{U}} \bigg[ \bigg\Vert \frac{\alpha_t}{K}   \sum^K_\{k=1\} \frac{1}{\mu} (f(w_t ,z_\{i_t\})- f(w'_t ,z_\{i_t\}))U^t_k \bigg\Vert \bigg] \leq \frac{L\alpha_t}{\mu} \Vert w_t -w'_t \Vert \mathbb{E}_\{U^t_1\} [\Vert U^t_1  \Vert ].$$
>
>  For small values of $\mu$ the one-point method is not stable, whereas ZoSS is stable for all $\mu>0$ (though at the expense of an additional -but crucial- "variance-reducing" function evaluation).

---

### Official Review · Reviewer_tUYw · 2022-06-30

**Rating:** 7
**Confidence:** 3
**Soundness:** 3 good
**Presentation:** 3 good
**Contribution:** 2 fair

**Summary:**

The paper establishes generalization error bounds of the gradient-free analogue of the SGD---zero-order stochastic search (ZoSS)---in which the gradient of the loss function at a point is approximated by $K + 1$ values ​​of the function in the vicinity of this point. The cases of nonconvex and bounded, convex and unbounded, nonconvex and unbounded loss functions are considered. However, in all cases, the authors require that the loss function be Lipschitz and smooth.

**Questions:**

How one shall choose the values of $K$ and $\mu$ in practice?

**Limitations:**

I think that the requirement of Lipschitz and smoothness is the main limitation of this work.

**Strengths And Weaknesses:**

**Strengths**

- The bounds are a natural generalization of the bounds by Hardt et al. (2016) to the case of ZoSS. It is noteworthy that the order of the convergence rate of the ZoSS is the same as that of the SGD.

- The proofs of the theorems are quite accessible: the reader only needs a college-level knowledge of Calculus, Linear Algebra, and Probability.

**Weaknesses**

- The requirement that the loss function be smooth calls into question the motivation for this work. After all, if we have smoothness and we know the loss function, then we can use the SGD. If we don't know the loss function, then how can we check its smoothness?

- The work lacks empirical validation of the theory on real data. It would also be great if the authors show a specific natural example where the ZoSS (under the paper's assumptions) is applicable and the SGD is not.

---

> ### Author Response · Authors · 2022-08-01
> **Response to Reviewer tUYw**
>
> We would like to thank the Reviewer for their effort reading our paper, and for their comments, which we address as follows.
>
> Q.1 ''The requirement that the loss function is smooth calls into question the motivation for this work. After all, if we have smoothness and we know the loss function, then we can use the SGD.''
>
> A.1 Recall, the algorithm does not require smoothness, the smoothness assumption is required to provide optimistic and meaningful generalization and excess risk guarantees. For the general non-smooth losses such optimistic guarantees are impossible even for the SGD algorithm, please also see the response to reviewer HnfQ for further details.
>
> Q.2 ''If we don't know the loss function, then how can we check its smoothness?''
>
> A.2 Given $K$ function evaluations we can estimate the smoothness parameter by estimating the (largest eigenvalue of the) Hessian. For Hessian estimation from Zeroth-Order function evaluations see Section 4.1 *''Estimating Hessian with Zeroth-Order Information''* in the paper *''Zeroth-Order Nonconvex Stochastic Optimization: Handling Constraints, High Dimensionality, and Saddle Points''* by  Balasubramanian \& Ghadimi. This technique shows how to estimate the smoothness parameter from function evaluations.
>
> Q.3 ''The work lacks empirical validation of the theory on real data. It would also be great if the authors show a specific natural example where the ZoSS is applicable and the SGD is not.''
>
> A.3 A standard, well-cited example of ZoSS application is adversarial learning as considered in [5], when the gradient is not known for the adversary (for additional applications for instance federated/reinforcement learning, linear quadratic regulators; please see also the introduction of the paper for references). Notice that the algorithm in [5] is restrictive in the high dimensional regime since it requires $2d$ function evaluations per iteration. In contrast, ZoSS can be considered with any $K\geq 2$ evaluation (the trade-off is accuracy vs resource allocation which is also controlled through $K$). If $K=d+1$ evaluation are available we recover guarantees of the deterministic zeroth-order approaches (similar to [5]). Experimental evaluation of the ZoSS algorithm on real data requires extensive cross-validation with training/test data-sets of varying sizes $n$, or even appropriate semi-synthetic data to effectively characterize the generalization error (and its decaying rate withe respect to $n$) through the experiment. Such experiments  are indeed interesting, but the focus of the current work is to provide rigorous theoretical guarantees in this setting.
>
> Q.4 ''How one shall choose the values of $K$ and $\mu$ in practice?''
>
> A.4 In practice $K$ is limited to the number of available function evaluations, it should be chosen as large as possible. $\mu $ should be small and satisfy the inequality $\mu\leq cL\Gamma^d_K/(n\beta (3+d)^{3/2}) $. Note that in practice it is often chosen between $10^{-10}$ and $10^{-8}$ (or even lower) and the ZoSS algorithm remains (numerically) stable (see [14]).

---

> > ### Comment · Reviewer_tUYw · 2022-08-03
> > **Thank you**
> >
> > Thank you for the detailed response. I am satisfied with your answers and will update the score accordingly.

---

### Official Review · Reviewer_HnfQ · 2022-07-10

**Rating:** 6
**Confidence:** 3
**Soundness:** 2 fair
**Presentation:** 3 good
**Contribution:** 3 good

**Summary:**


In this paper, the authors provide generalization error bounds for derivative-free updates under $L$-Lipschitz and $\beta$-smooth assumption of the loss fucntion.  The main technique extends  (Hardt et al.  2016) by bounding an additional gradient approximation error term ($\epsilon_{est}$).

**Questions:**



 Could the authors explain more about the proof of Lemma 2 to clarify my above concern? if $P(\mathcal{E}_t) = \frac{1}{n}$ instead of $P(\mathcal{E}_t) = 1- \frac{1}{n}$, how does this influence the generalization bounds?


**Strengths And Weaknesses:**


Pros.
1.  The generalization bounds for zeroth-order gradient update are new, and it may be potential for black-box learning tasks.

2.  The bounds are independent of the dimension $d$ and the number of function evaluations per step $K$.

Cons.

1.  In the proof of Lemma 2 (Line 194), it seems that $P(\mathcal{E}_t) = \frac{1}{n}$ instead of $P(\mathcal{E}_t) = 1- \frac{1}{n}$.
    If so, how does this influence the following proof and main theorem?

2. The $\beta$-smooth assumption of the loss function is a little bit strong. It requires the second-order derivative to be bounded.  It narrows down the potential loss functions because many practical loss functions may not have second-order derivatives, which makes the bounds less appealing.


===============================

Thanks for the authors' detailed response.  My main concern has been clarified.

---

> ### Author Response · Authors · 2022-08-01
> **Response to Reviewer HnfQ**
>
> We would like to thank the Reviewer for their effort reading our paper, and for their comments, which we address as follows.
>
>   Recall that $S,S'$ are two input sequences that differ in one exactly example, and $i_t \in\{1,2,\ldots,n\}$ is a random index chosen uniformly and independently by the random selection rule of the algorithm. The event $\mathcal{E}_t$ corresponds to the case $S(i_t)\equiv S'(i_t)$ (this also guarantees that $\tilde{G}_t (\cdot) \equiv \tilde{G}'_t (\cdot)$ at time $t$). Since $S ,S'$ have $n$ elements and they differ in one exactly entry it is true that $\mathbb{P} (\mathcal{E}_t )= 1 -1/n$ and $\mathbb{P} (\mathcal{E}^c_t )= 1/n$ (see also stability analysis for the SGD by Hardt et al. and other prior works).
>
>   We consider the smoothness assumption a possible limitation in the context of this paper, however it is known that optimization and generalization are often non compatible for non-smooth functions since in such cases extremely small step sizes are required to guarantee generalization. With such choices of step sizes (page 3, [45] ''Stability of Stochastic Gradient Descent
> on Nonsmooth Convex Losses'' by R. Bassily) the corresponding optimization error is large even for the SGD algorithm, thus small excess risk can not be guaranteed. As [45, page 3] showed for the SGD: *''Compared to the smooth case, the main difference is the presence of the additional $\eta \sqrt{T}$ term. This term has important implications for the generalization bounds derived from UAS. The first one is that the standard step size $\eta =\Theta (1/\sqrt{n})$ used in single pass SGD leads to a vacuous stability bound. Unfortunately, as shown by our lower bounds, this is unavoidable.''* Due to these inherent issues of non-smooth (and often convex only cases) we have left the ZoSS generalization error analysis on non-smooth losses for future work (please also see the last paragraph of our response to Reviewer Eioh above).

---

### Official Review · Reviewer_Eioh · 2022-07-11

**Rating:** 5
**Confidence:** 3
**Soundness:** 2 fair
**Presentation:** 3 good
**Contribution:** 2 fair

**Summary:**

The paper analyses the uniform stability of zero-order optimisation algorithms. Using the Gaussian randomisation for the gradient estimator that was put forward and extensively analysed by Nesterov and Spokoiniy, the authors establish the stability result in the spirit of Hardt et al [1] (who were only dealing with the pure SGD). This work focuses only on the generalisation bounds and ignores the optimisation error, referring to the work of Duchi et al [13].

**Questions:**

I addressed my main concerns in the previous part. I will update this part after the discussion phase and the authors' responses.

**Limitations:**

The authors addressed the limitations of their work. I will update this part after the discussion phase and the authors' responses.

**Strengths And Weaknesses:**

Overall the paper is relatively well written and easy to follow. My main concern is in the relevance and the novelty of the analysis. Nesterov and Spokoiniy [14] (see Theorem 1) show that the gradient estimator deployed by the authors of this submission is an *unbiased estimator* of a smoothed version of $f(\cdot, z)$. As long as the original function is Lipschitz (which is the case here), the smoothed version is a good approximation of the original one and the quality of this approximation is controlled by the parameter $\mu$. Thus, the algorithm analyzed by the authors of this submission is an SGD performed on a slightly different objective, which has identical regularity properties. Consequently, using the result of Hardt et al [1] in the context of SGD + the triangle inequality coupled with Theorem 1 from Nesterov and Spokoiniy [14], we immediately get the stability result for the zero-order variant considered in this submission (and it allows us to pick $\mu$ properly). Thus, I wonder if the authors could actually compact all their analysis into a couple of lines outsourcing most of the analysis from [1] and [14]?

What is actually pretty interesting is that if the original functions are NOT $\beta$-smooth, then the smoothed function has a Lipschitz gradient with the constant that depends on $\mu$. Hence, an interesting direction could be the exploration of the case of non-$\beta$-smooth loss functions. Unfortunately, this case is not considered by the authors.

---

> ### Author Response · Authors · 2022-08-01
> **Response to Reviewer Eioh**
>
> We would like to thank the Reviewer for their effort reading our paper, and for their comments, which we address as follows.
>
> The Reviewer suggests to replace the loss $f(w ,z_{i_t})$ with $f_\mu (w ,z_\{i_t\})= \mathbb{E}_U [ f(w+\mu U ,z_\{i_t\}) ]$, and then to apply the already existing results of Hardt et al. for the generalization error of the related smoothed surrogate. While at first glance this appears to be a reasonable approach to the problem (indeed, we had already considered this approach), below we explain why such an approach does not work. The Reviewer does refer to the smoothing parameter $\mu$, they do not refer to the number of function evaluations employed in the ZoSS update $K$. This is crucial.
>
> Of course, one can reformulate the generalization error in terms of $f_\mu$ as
>
> $$ \begin{align}
> &\mathbb{E}_\{S,A\} [ R(A(S)) -R_S (A(S))]  \\\\& = \mathbb{E}_\{S,A \} \left[ \mathbb{E}_z [f(A(S)),z] - \frac{1}{n}\sum^n_\{i=1\} f(A(S), z_i)\right] \\\\
> & = \mathbb{E}_\{S, A\} \bigg[  \mathbb{E}_z [ f(A(S),z)  -f_\mu (A(S),z)]\\\\  &\quad \quad\quad\quad-\frac{1}{n}\sum^n_\{i=1\} f(A(S),z_i) - f_\mu (A(S),z_i) + \mathbb{E}_z [ f_\mu (A(S),z)  ] -\frac{1}{n}\sum^n_\{i=1\} f_\mu (A(S),z_i) \bigg].
> \end{align} $$
>
> Then, as the Reviewer suggests, by applying the triangle inequality and Theorem 1 in [14], we find that \begin{align}
>     |\mathbb{E}_\{S,A\} [ R(A(S)) -R_S (A(S))]|   = \mathcal{O} (\mu \sqrt{d}) +
>     \bigg| \mathbb{E}_\{S,A\} \bigg[
>     \mathbb{E}_z [ f_\mu (A(S),z)  ] -\frac{1}{n}\sum^n_\{i=1\} f_\mu (A(S),z_i)\bigg]\bigg|,
> \end{align}
>
> and the second term on the right-hand side is the generalization error associated with the smoothed surrogate. As this decomposition holds for every algorithm, the question raised by the Reviewer is related to whether the decomposition is useful for reproducing the bounds developed in our paper just by applying the results of Hardt et al.. Let us explain why this is not quite correct.
>
> The issue is that *ZoSS is not an instance of such an SGD algorithm*. The reason for this is simple: *ZoSS does not have access to the "full-information" smoothed gradient* $\nabla f_\mu (w,z_{i_t})$, because in ZoSS the number of function evaluations $K$ is *finite* (also this is  the only implementable case). Recall that the ZoSS update is given as $W_\{t+1\} = W_t - \alpha_t {\Delta f}_ \{ w, z_\{i_t\}\}^\{K,\mu\} $, with ${\Delta f}_ \{ w, z_\{i_t\}\}^\{K,\mu\}= \frac{1}{K}\sum^K_\{k=1\}\frac{f(W_t +\mu U^t_\{k\},z_\{i_t\})-f(W_t ,z_\{i_t\} )}{\mu}U^t_\{k\}$ (this is also the case in [14]), and not $W_{t+1} = W_t - \alpha_t \nabla f_\mu (w , z_{i_t})$. When $K\rightarrow \infty$ these two gradient directions coincide, however for finite $K$, it is true that $\nabla f_\mu (w , z_\{i_t\}) \neq {\Delta f}_ \{ w, z_\{i_t\}\}^\{K,\mu\}$.
>
> Consequently, it follows that unless $K$ is infinite, the expression above is not useful/advantageous in the context of the ZoSS algorithm for providing generalization bounds. We can thus conclude that the comment
>
> *''Consequently, using the result of Hardt et al [1] in the context of SGD + the triangle inequality coupled with Theorem 1 from Nesterov and Spokoiniy [14], we immediately get the stability result for the zero-order variant considered in this submission (and it allows us to pick $\mu$ properly).''*
>
> is false, since the algorithm uses the direction ${\Delta f}_ \{ w, z_\{i_t\}\}^\{K,\mu\}$, and thus [14, Theorem 1] can not be considered directly in conjunction with the results of Hardt et al. (when $K$ is finite).
>
> To sum up, while the comment made by the reviewer is indeed correct for the case where $K=\infty$ (and only for that), our analysis applies to both cases of finite ($K<\infty$) and infinite (full-information regime) ($K\rightarrow\infty$) function evaluations. In practice only finite function evaluations are available.
>
> Finally, the reviewer correctly observes that $f_\mu$ is always $\beta_{f_\mu}$-smooth, even when the loss is non-smooth and $L$-Lipschitz. However, in such a case the smoothness parameter of $f_\mu$ is essentially unbounded as $\beta_{f_\mu}=\sqrt{d}L/\mu$ (Lemma 2 in [14]) and $\lim_{\mu \downarrow 0 } \beta_{f_\mu} \rightarrow \infty$. As a consequence, a potential direct variation of the stability analysis will provide pessimistic generalization error guarantees in contrast with the smooth loss case under consideration. Additionally, the lack of knowledge of the smoothed gradients (see above) does not allow using directly the smoothness of $ f_\mu$ (which is smooth even if $f$ is non-smooth) in a direct way and introduces further artifacts and instability terms that appear in the analysis. Nonetheless, we believe that studying generalization in the context of nonsmooth losses using smooth approximation does seem a very interesting direction for developing further technical approaches/results (we thank the Reviewer for that point); however, this is out of the scope of the present paper.

---

> > ### Comment · Reviewer_Eioh · 2022-08-08
> > **Aknowledging**
> >
> > I would like to thank the authors for their response. After reading the rebuttal and the other reviews, I wish to modify my review and increase my score to a 5.

---

### Author Response · Authors · 2022-08-01
**Generic Response**

We would like to thank the reviewers for their detailed comments on our paper. We believe that the feedback and the subsequent revision will greatly improve the manuscript. All typos and minor comments will be addressed accordingly (for instance line 43, comment by Reviewer HbvR). We are planning to address all comments as best as possible, either in the main body of the paper, or in the supplement, in a revised version after the reviewing process. We continue by providing our responses to each reviewer.

---

### Meta-Review · Area_Chair_J8MY · 2022-08-26

**Recommendation:** Accept
**Confidence:** Less certain

**Metareview:**

The paper provides generalization bounds for zeroth order stochastic search (ZoSS) based on algorithmic stability. The paper appears to follow from a fairly modest modification of Hardt, Recht and Singer `16, but the consensus among reviewers is that this modification is not trivial, and the bounds are novel and interesting. Consequently, I recommend acceptance.

**Award:**

No

---

### Decision · Program_Chairs · 2022-09-14

Accept